# The venom gland transcriptome of *Tityus paraguayensis* reveals a diverse array of bioactive molecules from the Brazilian Cerrado

Henrique Ranieri Covali-Pontes[1], Brayhan Meneguelli[1], Jéssica de Moraes Carretone[1], Alynne Coelho Ribeiro[1], Angélica Camargo dos Santos[2], Thais Fernanda Carlos[2], Marcos Roberto Chiaratti[2], Milene Ferro[3], Flávio Henrique Silva[2], Renata dos Santos Rodrigues[4], Malson Neilson Lucena[1]*

1 Institute of Biosciences, Federal University of Mato Grosso do Sul, Campo Grande, Mato Grosso do Sul, Brazil, 2 Department of Genetics and Evolution, Federal University of São Carlos, São Carlos, Brazil, 3 Department of General and Applied Biology, Paulista State University, Rio Claro, Brazil, 4 Institute of Biotechnology, Federal University of Uberlândia, Uberlândia, Brazil

* malson.lucena@ufms.br

## Abstract

Scorpions are arthropods with venom glands at their telson that produce chemicals such as peptides and proteins. These compounds may have pharmacological effects, including antimicrobial, ion channel modulating, and antihypertensive activities. Our study aims to examine the transcripts from the venom glands of *Tityus paraguayensis*, focusing on identifying and annotating the genes expressed in these glands. A transcript encoding a potassium channel-modulating peptide was selected for 3D structural modeling, phylogenetic analysis, and interaction assessment. Initially, the scorpions' telsons were dissected and analyzed using transcriptome sequencing. The data were then assembled and functionally annotated. The sequencing and assembly of the venom gland transcriptome produced a set of 37,283 transcripts, of which 523 were annotated as potentially related to venom components. Among the venom components, peptides that modulate sodium (8%), potassium (9%), and calcium (1%) channels, antimicrobial peptides (6%), antihypertensives (2%), phospholipases (1%), and metalloproteinases (29%) were identified, along with other compounds (44%). Specific highlights include the structural-functional analysis of four key peptides: TpNa3, a probable β-toxin sodium channel modulator with a βαββ structural motif; TpHyp1, a long-chain antihypertensive peptide that contains the conserved KPP motif; TpAP1, a short antimicrobial peptide with a low positive charge and an α-helical structure; and TpK8 is a potassium toxin that was previously partly identified (α-KTx). Molecular modeling and docking analyses showed that TpK8 binds with high affinity and stability, especially to the Kv1.3 channel, through specific interactions with the selectivity filter. These findings emphasize the extensive molecular diversity of *T. paraguayensis* venom, highlighting its potential as a rich and largely unexplored

**Data availability statement:** All relevant data are within the manuscript and its Supporting information file.

**Funding:** MRC FAPESP 23/17188-4; FHS FAPESP 23/17182-6 ACS FAPESP 22/15264-2 MNL This work was conducted with support from the Federal University of Mato Grosso do Sul—UFMS/MEC—Brazil. This study was also financed in part by the Coordenac.o de Aperfeicoamento de Pessoal de Nivel Superior—Brasil (CAPES)—Finance Code 001. The funders had no role in study design, data collection and analysis, decision to publish, or preparation of the manuscript. There was no additional external funding received for this study.

**Competing interests:** The authors have declared that no competing interests exist.

source of bioactive molecules. This makes it a promising target for developing new bioactive compounds for biotechnological and therapeutic use.

## Introduction

Scorpions are venomous arthropods in the class Arachnida and the order Scorpiones, which includes 24 families, 247 genera, and around 2,903 species, most of which have been found in tropical and subtropical regions [1–3]. They are among the oldest arthropods, dating back to 437.5–436.5 million years ago [3]. Scorpions have a broad geographic range, living on every continent except the poles [4]. Their adaptable diversity allows them to live in environments from deserts to humid caves, showing specialized features and strategies for occupying various ecological niches [5].

Scorpion stings occur when these animals use their venom as a defense. If not treated, such stings can sometimes be deadly, posing a neglected public health issue in tropical and subtropical countries, where there is greater species diversity [6,7]. In Brazil, scorpion envenomation continues to be a public health concern, with an increase in reported cases in recent years, totaling more than 200,000 scorpion stings officially recorded in 2023 [8]. Most scorpion stings (about 95%) lead to mild or no symptoms. Systemic effects of envenomation only occur when venom levels reach a critical plasma point, which varies based on both scorpion and victim traits [6].

The components in scorpion venom can be categorized into two groups: non-protein molecules (such as mucus, polysaccharides, lipids, nucleotides, inorganic salts, and metals) and protein molecules (including peptides, enzymes, and non-enzymatic proteins), with peptides and enzymes being the most common [9]. Peptides make up a large part of scorpion venoms and can be classified based on their structure, targets, pharmacological activity, and other factors [10]. Peptides are divided into two main groups: disulfide-bridged peptides (DBPs) and non-disulfide-bridged peptides (NDBPs) [11]. Although about 3,000 DBPs and 200 NDBPs have been identified, it is estimated that these account for less than 2% of the total peptides found in scorpion venoms [12]. The primary enzymes in scorpion venom include metalloproteinases, serine proteases, and cysteine proteases. Additionally, proteomes and transcriptomes have revealed other enzymes, including phospholipases, hyaluronidases, nucleotidases, lipases, chitinases, and amylases [13].

*Tityus paraguayensis* (Fig 1) is found in the state of Mato Grosso do Sul and is also found in Paraguay and northern Argentina. These scorpions are about 28–35 mm long and have a dark, variegated coloration across their bodies [14], and they possess a diploid karyotype (2n) with 16–18 chromosomes [15]. A recent study, using electrophoretic and chromatographic analyses, revealed that the venom of *T. paraguayensis* contains a wide range of compounds. Additionally, shotgun proteomics partially identified five potassium channel toxins, three sodium channel toxins, and one metalloproteinase [16].

Because of the diverse compounds in scorpion venoms, they are considered promising sources for the discovery of molecules with potential pharmacological applications [17]. Finding new molecules provides an essential alternative

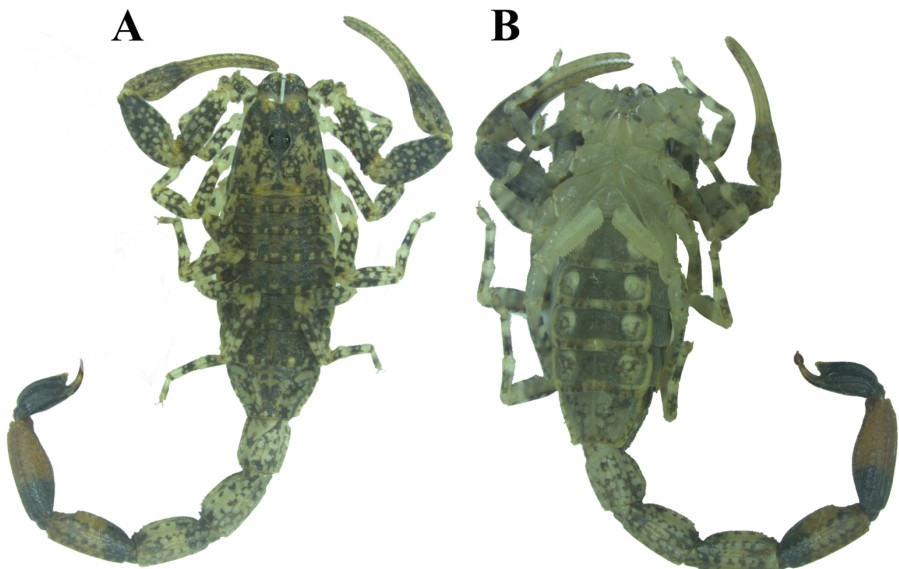

**Fig 1.** *Tityus paraguayensis* scorpion, A – dorsal view; B – ventral view.

for addressing public health challenges faced by humanity, as well as the therapeutic difficulties related to emerging, re-emerging, and neglected infectious diseases [18]. In addition, high mortality rates continue to be associated with bacterial resistance to current antibiotics and cancer-related deaths [19,20].

Animal venoms have been extensively studied for bioprospecting molecules. Techniques such as "omics" approaches enable comprehensive analysis of biomolecules within organisms, tissues, or cells. These "omics" sciences have advanced, delivering better results at lower cost [21]. Among the various "omics" techniques, genomics, transcriptomics, proteomics, and metabolomics are the most widely used. However, other approaches include epigenomics, interactomics, and lipidomics [22]. More than 37 scorpion species have had their venom glands analyzed using transcriptomics, with *Tityus serrulatus* standing out for having 233 compounds identified in its venom [13,22].

This study analyzed the transcriptome of *T. paraguayensis* venom glands, providing a comprehensive profile of their expressed toxin repertoire. Most of the identified sequences were putative peptide-encoding transcripts, and several were further analyzed for their physicochemical properties and similarity to known venom components. Additionally, a specific transcript encoding a potassium channel-modulating peptide—whose amino acid sequence had been partially identified in earlier proteomic studies [16]—was chosen for detailed three-dimensional structural modeling, phylogenetic analysis, and assessment of its potential interactions with potassium channels.

We identified new neurotoxins, enzymes, and other venom components, many of which resemble known scorpion toxins, highlighting their evolutionary importance. Notably, we found peptides from the hypotensin venom family, confirming their presence in Buthidae. To our knowledge, this is the first transcriptomic analysis of *T. paraguayensis* venom glands, providing valuable insights into its venom's composition and therapeutic potential and deepening our understanding of its venom.

## Materials and methods

### Collection and storage of specimens

Specimens of *T. paraguayensis* (SISGEN: AE2EA94) were collected from the Private Reserve of Natural Heritage of the Federal University of Mato Grosso do Sul (RPPN-UFMS) in Campo Grande, Mato Grosso do Sul, Brazil (20°30'25.6"S,

54°37'1.2"W), under environmental license No. 86953−1. Voucher specimens were identified and deposited in the Zoological Collection of UFMS (ZUFMS-CHE00534). Collections occurred in October 2022 during the evening (7:00 PM–9:00 PM), using active searching with LED-UV 395 nm flashlights. After collection, scorpions were kept individually in plastic containers with water available at all times and fed weekly with *Drosophila melanogaster*.

### RNA extraction

Total RNA was isolated using the PicoPure® RNA Isolation Kit (Thermo Fisher Scientific) according to the manufacturer's protocol. Briefly, telsons from two specimens were homogenized in 50 µL of extraction buffer and incubated at 42°C for 30 minutes. The sample pool was centrifuged at 800 × g for 2 minutes, and the supernatant was mixed with 70% ethanol before being applied to the purification column. Successive washes were performed, and RNA was eluted in 30 µL of elution buffer through sequential centrifugation (1 min at 1,000 × g and 1 min at 16,000 × g). RNA quantity and purity were measured using a Nanodrop ND-1000 spectrophotometer (Thermo Scientific). Samples were immediately stored at –80°C. RNA integrity was assessed by capillary electrophoresis using the Agilent 2100 Bioanalyzer®.

### cDNA library construction and sequencing

cDNA libraries were prepared using the TruSeq® mRNA kit (Illumina). Poly(A)+ mRNA was isolated with oligo(dT) magnetic beads, fragmented at high temperature with divalent cations, and reverse-transcribed using random hexamers. Second-strand cDNA was synthesized with DNA polymerase I, and RNA was degraded with RNase H. Sequencing adapters were attached, and libraries were amplified through PCR. The libraries were sequenced on a NextSeq 550 platform (Illumina®) with paired-end reads. Raw transcriptomic data from *T. paraguayensis* venom glands were uploaded to NCBI under BioProject PRJNA1268009, BioSample SAMN48739814, and SRA SRR33699524.

### Quality control, preprocessing, *de novo* assembly, and functional annotation

Quality control of the reads was performed using FastQC (Babraham Bioinformatics; www.bioinformatics.babraham.ac.uk/projects/fastqc/). FASTX Toolkit (http://hannonlab.cshl.edu/fastx_toolkit/) was used to remove the first 15 bp and the last 4 bp, aiming to solve the GC bias problem [23,24]. Transcriptome assembly was performed *de novo* with Trinity v2.3.2 [25] with a minimum contig length of 200 bp. Assembly completeness was evaluated with BUSCO v5 [26], using the Metazoa database. Functional annotation was performed in OmicsBox v3.0.25 (BioBam, Valencia, Spain) using BlastX [27,28] with an E-value cutoff of 1E-05 against the Metazoa NCBI database. Functional annotation based on Gene Ontology (GO) was performed across the three main categories: Biological Process (BP), Molecular Function (MF), and Cellular Component (CC). Protein domains were identified using InterProScan [29], and corresponding Enzyme Commission (EC) numbers were assigned. Additionally, Kyoto Encyclopedia of Genes and Genomes (KEGG) annotation was conducted to identify metabolic and signaling pathways associated with the differentially expressed genes (DEGs).

Transcripts classified as venom-related were manually curated based on subcellular localization and previous descriptions in scorpions, using the keywords "toxin," "venom," and "extracellular region." Transcript abundance was estimated as TPM (Transcripts Per Million) using RSEM [28].

### In silico peptide analyses

Open reading frames (ORFs) were predicted with TransDecoder v5.5.0 [30] in OmicsBox v3.0.25. Signal peptides and propeptide regions were identified using SignalP 6.0 (https://services.healthtech.dtu.dk/services/SignalP-6.0/) and ProP 1.0 (https://services.healthtech.dtu.dk/services/ProP-1.0/), respectively. Physicochemical parameters were measured using PepCal (https://pepcalc.com/). The MEGA.12 software (https://www.megasoftware.net/) was used to perform alignments with the ClustalW algorithm [31]. Structural modeling of peptides was carried out with AlphaFold [32].

## Phylogenetic analysis

The phylogenetic analysis aimed to clarify the evolutionary relationships of the potassium channel toxin (TpK8) from *T paraguayensis*. Proteins similar to TpK8 were identified using BLASTp (https://blast.ncbi.nlm.nih.gov/Blast.cgi), excluding synthetic peptides, resulting in 42 sequences. An outgroup was included, consisting of the toxin Hanatoxin from *Grammostola spatulata*. The peptide sequences were aligned using MEGA12 with the ClustalW method. A Maximum Likelihood phylogenetic tree was then built with 1000 bootstrap replicates and without excluding any sequences.

## Molecular docking

The three-dimensional structures of potassium channels Kv1.2, Kv1.3, and Shaker were retrieved from the Protein Data Bank, with the respective IDs 2A79, 7EJ1, and 7SIP. In PyMOL, these structures were processed by removing water molecules, crystallographic ligands, and extraneous chains. Only the subunit containing the selectivity filter (TVGYG) was kept for docking.

Molecular docking between the peptide and each channel was conducted using HADDOCK 2.4 (High Ambiguity Driven protein–protein DOCKing). Active residues in the peptide were identified based on previously known structural data for K$^+$ channel-blocking toxins. For the channels, active and passive residues were selected from those within 5 Å of the extracellular pore and the selectivity filter. Docking was performed in the standard protein–peptide mode, resulting in multiple clusters. The top cluster, based on the HADDOCK score, was chosen for detailed structural analysis.

Intermolecular interactions were examined using two complementary approaches: LigPlot+ created 2D diagrams of hydrophobic contacts and hydrogen bonds, while PyMOL offered detailed 3D analysis of the interface, emphasizing pore-blocking geometry and producing the final images displayed in the results.

# Results

### RNA extraction, quality control, preprocessing, and *de novo* assembly

A total of 30 μL of RNA was recovered at a concentration of 2.3 ng/μL. Capillary electrophoresis revealed high RNA integrity, with minimal degradation and a distinct 18S rRNA peak around 40 seconds (S1 Fig). The absence of the 28S rRNA peak prevented the calculation of RIN. A total of 49,436,548 reads were generated. FASTX Toolkit removed the first 15 bp and the last 4 bp to address GC bias (S2 and S3 Figs). Performing *de novo* assembly with Trinity 37,283 transcripts and 29,304 genes with a GC content of 35.69 were identified. In addition, an N50 value of 1,213 was obtained, and a 787.9 mean length of transcripts (bp). BUSCO analysis revealed the completeness and integrity of the assembled transcriptome against the Metazoa ortholog bank, indicating that 75.68% of the genes were complete (single-copy and duplicated).

### Transcriptome overview

Functional annotation was performed for 37,283 assembled transcripts using the OmicBox platform. Out of 37,283 assembled transcripts, 23,989 were annotated (64.34%). Among these, 23,466 (62.94%) were classified as related to glandular tissue, and 523 (1.4%) as potential venom components (Fig 2). Venom-related transcripts included ion channel modulators (sodium, potassium, and calcium), antimicrobial peptides, antihypertensive peptides, metalloproteases, phospholipases, and other components. Although metalloproteases made up 29% of the diversity in venom-related transcripts, they only accounted for 8% of the abundance in TPM (Fig 3). Conversely, ion channel modulators, antimicrobial peptides, and antihypertensive peptides appeared more frequently than suggested by their transcript diversity. Phospholipases were scarcely expressed (0.0001% of venom). Additionally, transcripts like enzymes, allergens, protease inhibitors, uncharacterized peptides, and hypothetical secreted proteins (S1 Table) exhibited low abundance despite their diversity.

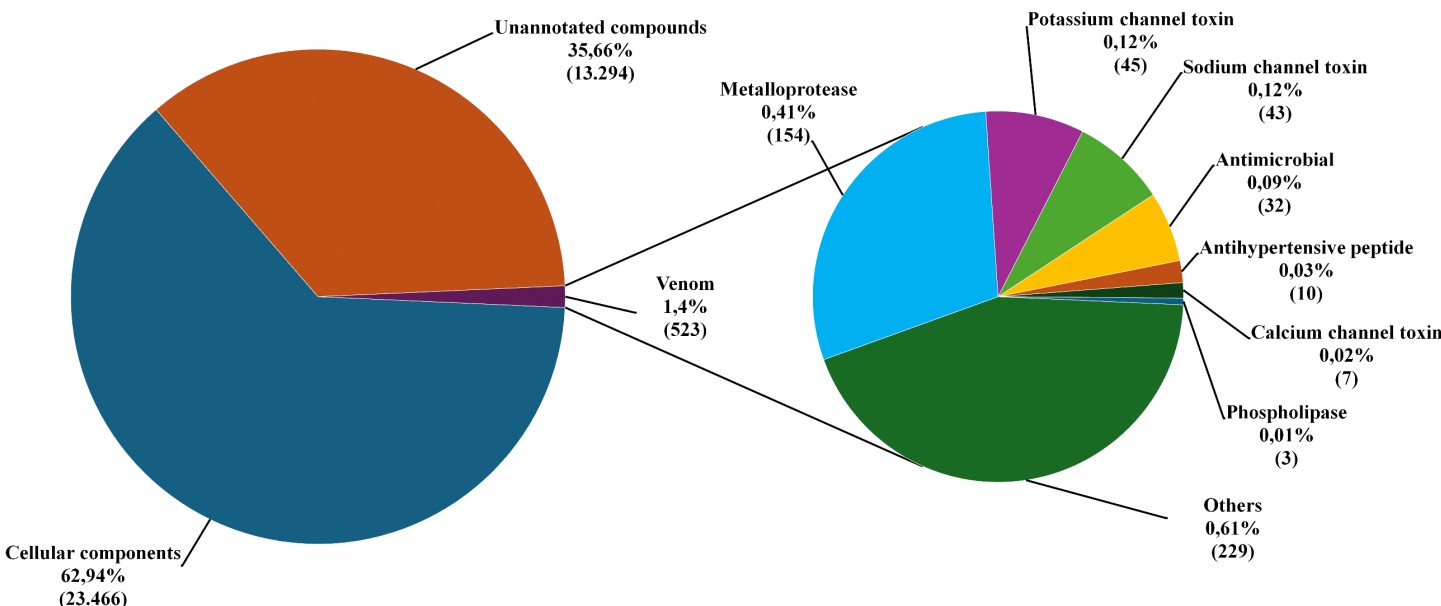

**Fig 2. Relative distribution of transcripts identified in the venom glands of *Tityus paraguayensis*, based on transcriptome functional annotation.**

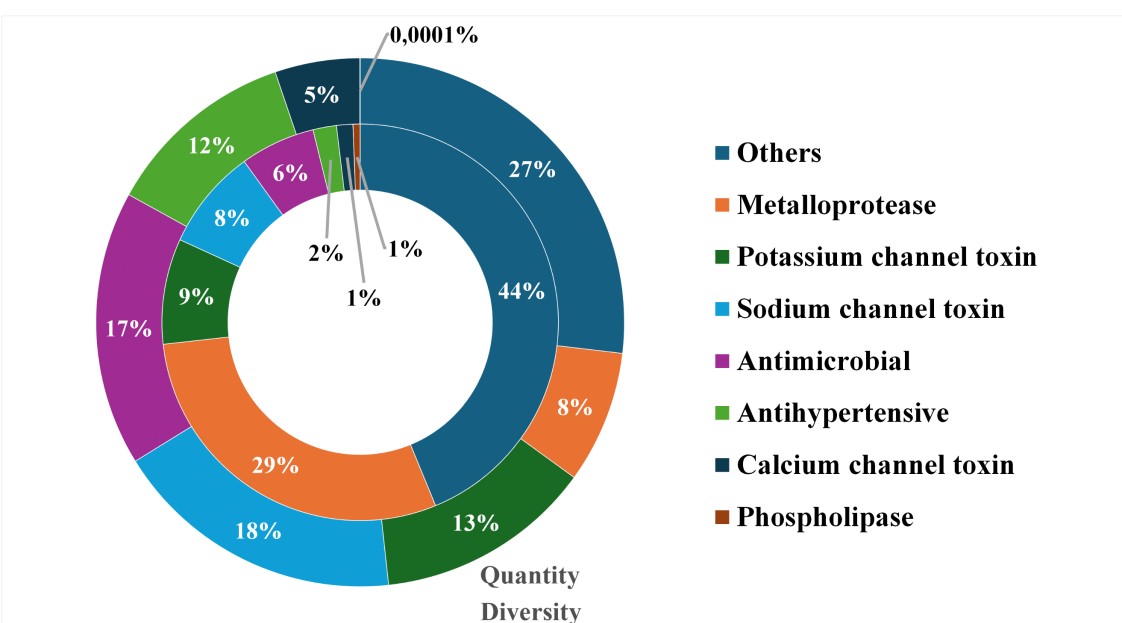

**Fig 3. Relative distribution of toxins in *Tityus paraguayensis* venom, represented by "diversity" (inner ring) and "abundance" (outer ring).** Diversity was measured as the number of distinct transcripts identified for each compound class, while abundance was calculated as the sum of each class's transcript TPM values.

Out of the 30 most expressed transcripts, 16 were identified as venom components, including two antimicrobial peptides, three antihypertensive peptides, five ion channel modulators, and six other venom proteins (Table 1). Additionally, unannotated transcripts, hypothetical proteins, and genes related to glandular tissue were detected. Since the transcripts related to venom peptides were the most abundant after functional annotation, one representative from each of the four main classes will be detailed in the following sections.

**Table 1. The top 30 transcripts with the highest expression levels in the venom gland transcriptome of *Tityus paraguayensis*, ranked by TPM values. Transcripts shown in bold are putative venom components.**

| ID do transcrito | Descrição | Organismo | NCBI | Comprimento (bp) | TPM |
|---|---|---|---|---|---|
| TRINITY_DN17215_c6_g1_i1 | Hypothetical protein, partial | | | 1195 | 188872.4 |
| TRINITY_DN8073_c1_g1_i1 | No hits found | | | 213 | 47368,.6 |
| TRINITY_DN12769_c0_g1_i1 | **Non-disulfide-bridged peptide 4.23-like; NDBP-4.23** | *Tityus costatus* | Q5G8B5.1 | 356 | 18904,.4 |
| TRINITY_DN6455_c0_g1_i1 | **Anionic peptide** | *Centruroides sculpturatus* | XP_023227050.1 | 390 | 18706,.9 |
| TRINITY_DN10353_c0_g1_i5 | **Beta-mammal Tt1g; T.trivittatus toxin 1 gamma-like** | *Tityus trivittatus* | P0DMM8. 1 | 412 | 10863.81 |
| TRINITY_DN4868_c1_g1_i2 | **Anti-hypertensive peptide; Hypotensin toxin Ts14.1;** | *Tityus serrulatus* | P84189.2 | 418 | 10358.58 |
| TRINITY_DN17215_c7_g1_i1 | No hits found | | | 204 | 9716.85 |
| TRINITY_DN14241_c1_g1_i1 | **Calcium channel toxin-like peptide-1** | *Tityus serrulatus* | P0DM29.1 | 613 | 9595.8 |
| TRINITY_DN21617_c1_g1_i1 | Muscle LIM protein | *Centruroides sculpturatus* | XP_023242286.1 | 697 | 8975.55 |
| TRINITY_DN3600_c0_g1_i1 | No hits found | | | 569 | 8549.55 |
| TRINITY_DN2762_c0_g1_i4 | No hits found | | | 237 | 7306.94 |
| TRINITY_DN19411_c0_g1_i1 | **Bradykinin-potentiating peptide** | *Tityus discrepans* | C9X4J0.1 | 587 | 6846.98 |
| TRINITY_DN3918_c0_g1_i1 | **Venom protein VP6** | *Centruroides sculpturatus* | XP_023218944.1 | 977 | 6632.17 |
| TRINITY_DN2805_c0_g1_i1 | **Non-disulfide-bridged peptide androcin 18−1** | *Androctonus bicolor* | AIX87617.1 | 695 | 6577.64 |
| TRINITY_DN1521_c1_g1_i7 | **Potassium channel toxin epsilon-KTx 1.1; Tityustoxin-11** | *Tityus serrulatus* | P0C174.2 | 316 | 5823.46 |
| TRINITY_DN8517_c0_g1_i2 | **Sodium beta toxin** | | | 585 | 5750.49 |
| TRINITY_DN13858_c0_g1_i2 | **Venom peptide meuPep31** | *Mesobuthus eupeus* | AMX81469.1 | 461 | 5463.89 |
| TRINITY_DN687_c1_g1_i13 | Hypothetical protein, partial | *Tityus discrepans* | CAY61861.1 | 529 | 5012.13 |
| TRINITY_DN93_c0_g1_i1 | No hits found | | | 363 | 4829.75 |
| TRINITY_DN201_c0_g1_i1 | Myosin regulatory light chain 2 | *Centruroides sculpturatus* | XP_023218181.1 | 1104 | 4814.23 |
| TRINITY_DN2489_c0_g1_i2 | **Toxin Tf2;** | *Tityus fasciolatus* | C0HJM9.1 | 494 | 4813.61 |
| TRINITY_DN1169_c0_g1_i5 | **Potassium channel blocker AbKTx-5** | *Androctonus bicolor* | AIX87694.1 | 809 | 4802.97 |
| TRINITY_DN1073_c0_g1_i1 | cytochrome c oxidase subunit II (mitochondrion) | *Centruroides vittatus* | YP_009470513.1 | 2944 | 4587.77 |
| TRINITY_DN18308_c0_g1_i1 | hypothetical protein, partial | *Tityus discrepans* | CAY61876.1 | 536 | 4583.57 |
| TRINITY_DN13601_c0_g1_i4 | CG9896-like protein | *Centruroides sculpturatus* | XP_023218950.1 | 964 | 4331.14 |
| TRINITY_DN2371_c0_g1_i1 | Hypothetical protein | *Tityus discrepans* | CAY61897.1 | 513 | 4140.43 |
| TRINITY_DN10847_c0_g1_i1 | **Scorpine-like** | *Tityus costatus* | Q5G8A6.1 | 899 | 4121.18 |
| TRINITY_DN10711_c0_g1_i1 | Cytochrome c oxidase subunit I | *Tityus clathratus* | AAU07854.1 | 2001 | 3830.56 |
| TRINITY_DN1250_c0_g1_i3 | **Venom protein AbVp-9** | *Androctonus bicolor* | AIX87809.1 | 1000 | 3726.59 |
| TRINITY_DN4868_c1_g1_i1 | **Anti-hypertensive peptide; Tityustoxin-14 1;** | *Tityus serrulatus* | P84189.2 | 448 | 3561.79 |

## Sodium channel modulator peptides

Among the 43 transcripts identified within the sodium channel modulator peptide class, the two with the highest TPM values lacked complete open reading frames (ORFs) upon translation. Therefore, subsequent analyses used the peptide with the third-highest TPM in this class, called TpNa3. This peptide contains a 19-amino-acid signal sequence and a 65-residue mature peptide. Additionally, eight cysteine residues were identified, which form four disulfide bonds (Table 2). TpNa3 has a theoretical molecular mass of 7303.21 Da, an isoelectric point of pH 9.32, a net charge of +2.2 at pH 7, and is highly soluble in water.

The sodium channel modulator TpNa3 showed the highest similarity when aligned with To12 (H1ZZI1.1) from *Tityus obscurus*, Ts2 (P68410.2) from *T. serrulatus*, and Tf2 (C0HJM9.1) from *Tityus fasciolatus*, with an identity percentage of 82.8%. Additionally, peptides from *Tityus stigmurus* (Tst1 (P68411.1) and Tst2 (P56612.1)), *Tityus bahiensis* (Tb2 (P56609.1) and Tb2-II (P60276.1)), and *Tityus melici* (NaTx-_Tm1 (WLF82730.1), 2 (WLF82731.1), and 3 (WLF82729.1)) exhibited identity percentages ranging from 70.3% to 82.3% with TpNa3. Conserved regions are observed across all peptides, with cysteine residues consistently conserved (Fig 4).

In silico analysis of TpNa3 revealed a structure featuring a right-handed α-helix from residue 23 (phenylalanine) to 29 (cysteine). It also included an antiparallel β-sheet composed of three β-strands: β1 (residues 2–4), β2 (36–40), and β3 (43–49). The structure showed random coil regions in the carboxy-terminal segment (50–65) and in the connecting segment between the α-helix and β1 (5–22) (see Fig 5). Predicted disulfide bonds form between cysteine residues at positions 24–44, 16–39, 28–46, and 12–63, with bond lengths around 2.0 Å. Bonds 24–44 and 28–46 link the α-helix to β3, while the 16–39 bond connects β2 to a random coil segment. The 12–63 bond joins two random coil regions.

**Table 2. Sequence and TPM of peptide TpNa3. The bolded region indicates the signal peptide, and the shaded residues represent cysteine amino acids.**

| Peptide | Sequence | TPM |
|---|---|---|
| TpNa3 | **MKELLFFISIFVMIEIIAA**TKEGYPMDHKGCKYSCAVRPSGFC DRYCKLHLSAGSGYCAWPACYCYGVPDDKPLWDYDTNKCGK | 4.813,61 |

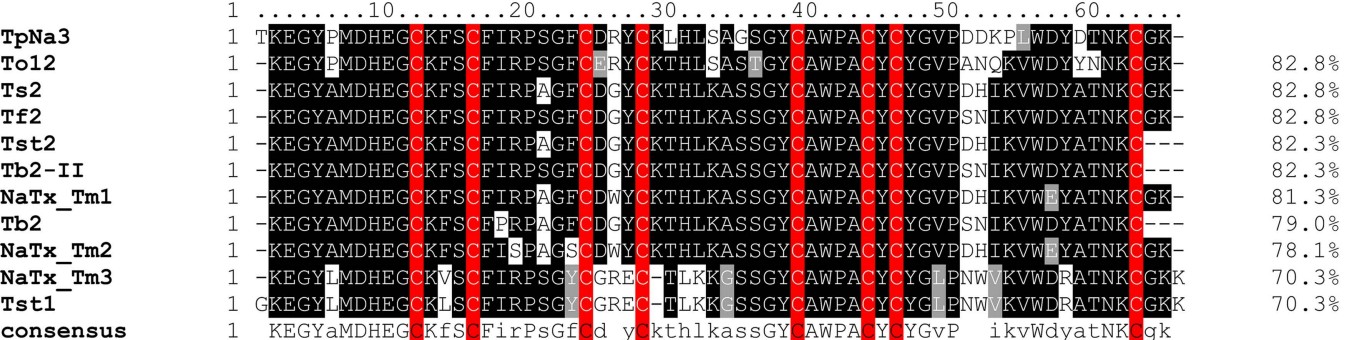

**Fig 4. Alignment of amino acid residues of TpNa3 found in the venom gland transcriptome of *Tityus paraguayensis* with Ts2 (P68410.2) from *Tityus serrulatus*, NaTx Tm1, Tm2, and Tm3 (WLF82730.1, WLF82731.1, and WLF82729.1) from *Tityus melici*, Tst1 and Tst2 (P68411.1 and P56612.1) from *Tityus stigmurus*, Tb2 and Tb2-II (P56609.1 and P60276.1) from *Tityus bahiensis*, Tf2 (C0HJM9.1) from *Tityus fasciolatus*, and To12 (H1ZZI1.1) from *Tityus obscurus*.** Gray scale variations indicate sequence conservation. Letters in uppercase show 100% conserved amino acids, lowercase letters mark the most conserved residues, blank spaces indicate highly variable positions with no dominant amino acid, and dashes (–) represent gaps inserted to maintain homology, reflecting insertions or deletions. The percentage values to the right indicate the similarity between TpNa3 of *Tityus paraguayensis* and the other sequences.

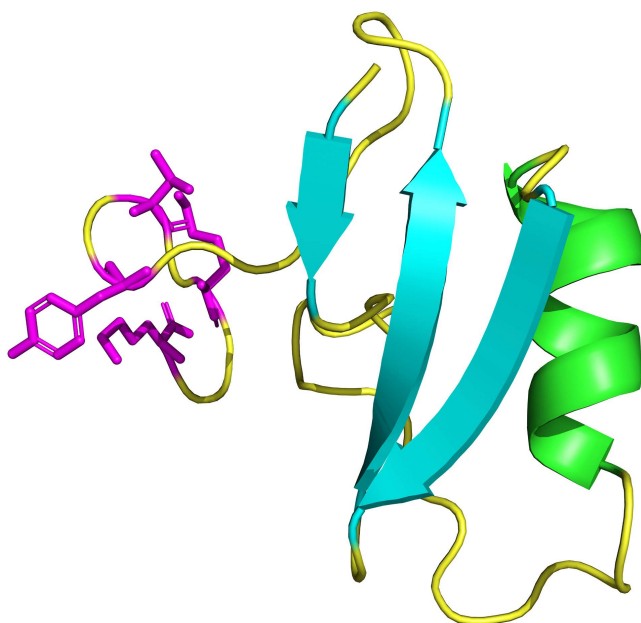

**Fig 5. Predicted three-dimensional structure of the TpNa3 peptide, generated through structural modeling with the AlphaFold server.** The image was rendered using PyMOL for clarity. The protein features a structure with β sheets (cyan) and a distinct α helix (green), linked by flexible loops (yellow), characteristic of NaTx toxins. At the C-terminal end, three key residues—lysine (Lys), tyrosine (Tyr), and tryptophan (Trp)—are highlighted in magenta.

## Antihypertensive peptides

Ten peptides were identified as potentially belonging to the antihypertensive or hypotensin group. The peptide with the highest TPM features a 35-amino-acid signal sequence and an estimated mature peptide consisting of 49 residues (see Table 3). This molecule has a molecular mass of 5379.85 Da, an isoelectric point of pH 4.12, a net charge of −5, and is highly water-soluble.

Sequence alignment of TpHyp1 shows that the central region, between residues 13 and 37, has complete coverage with the mature peptide TistH (P0DY04.1) from *T. stigmurus*, Ts31 (A0A218QWY8.1), and TsHpt-1 (P84189.2) from *T. serrulatus* (Fig 6).

According to AlphaFold predictions, TpHyp1 forms a right-handed α-helix spanning residues 22–31. Both the N- and C-terminal regions are composed of random coil segments without a defined secondary structure (Fig 7). No disulfide bond was detected in the structure. The molecule's core contains the KPP domain, composed of lysine-proline-proline residues, which creates a rigid structure characteristic of scorpion hypotensins. This KPP motif helps stabilize the overall shape and modulates how the molecule interacts with its targets. Located between the flexible N-terminal region and the C-terminal helix, it is crucial in determining the overall conformation of the peptide, thereby impacting its biological function and specificity.

**Table 3. Sequence and TPM of the TpHyp1 peptide, with the signal peptide highlighted in bold.**

| Name | Sequence | TPM |
|------|----------|-----|
| TpHyp1 | **TSNYSNISKTNIKMKMMIPVLLCILLLMHSLSSTA**TSLEDE-QQSVKRGAPDFTGIAADIIKKIKETGAKPPARFDSEAFEIEED | 10.358,58 |

```
                              1 ........10........20........30........40........50
TpHyp1                        1 TSLEDEQQS-VKRGAPDFTGIAADIIKKIKETGAKPPARFDSEAFEIEED
TistH                         1 ------------ADMDFTGIAESIIKKIKETNAKPPA-----------  76.0%
Ts31                          1 ------------ADVDFTGIADSIIKKIKETNAKPPA-----------  76.0%
TsHPT-I                       1 ------------AEIDFSGIPEDIIKQIKETNAKPPA-----------  68.0%
Tm_Putative_hypotensin_(1)    1 --LEDEQENMEERAEVDFSGIPEDIIKQIKETNAKPPARFDPAAFEKS--  63.0%
Tm_Putative_hypotensin_(2)    1 -METEQQNMEERADVDFTGIAESIIKKIKETNAKPPARFDPATFGENED   62.5%
consensus                     1  le eq     radvDFtGIae IIKkIKETnAKPPArfd    f
```

Fig 6. Alignment of the putative hypotensin TpHyp1 identified in the venom gland transcriptome of *Tityus paraguayensis* with other hypotensins previously described in different scorpion species: Tm putative hypotensin (1) and Tm putative hypotensin (2) from *Tityus melici* (WLF82706.1 and WLF82708.1), TistH from *Tityus stigmurus* (P0DY04.1), and two hypotensins from *Tityus serrulatus*: TsHpt-I (P84189.2) and Ts31 (A0A218QWY8.1). Gray scale variations depict sequence conservation. Uppercase letters in the consensus line indicate 100% conservation, lowercase letters show the most conserved residues, blank spaces mark highly variable positions, and dashes (–) represent alignment gaps. The percentage values on the right show the similarity between TpHyp1 of *Tityus paraguayensis* and other sequences.

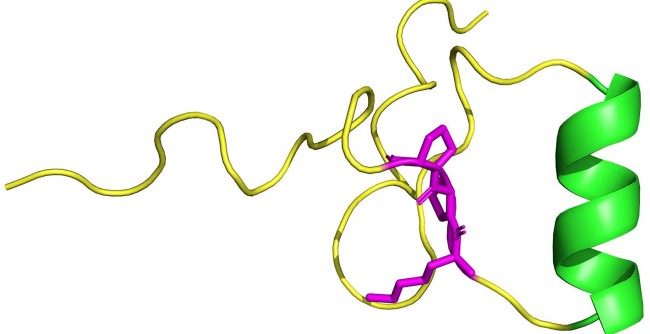

Fig 7. Predicted three-dimensional structure of the TpHyp1 peptide, generated through structural modeling with the AlphaFold server. The image was visualized using PyMOL for illustration. The alpha helix (green) is at the far right of the molecule. Flexible loops (yellow) and functional residues (magenta) in the KPP domain, linked to hypotensive effects, are also present.

## Antimicrobial peptides

A total of 33 potential antimicrobial compounds were identified. The peptide with the highest TPM was translated and analyzed to determine its chemical and functional properties. This peptide comprised a signal sequence and a propeptide region (see Table 4), and was designated TpAP1. The conserved GRXXR motif was identified, suggesting potential C-terminal amidation. Following post-translational modifications, a mature peptide consisting of 17 amino acids was generated. TpAP1 has a molecular mass of 1699.15 Da, a net charge of +1 at pH 7, and exhibits low water solubility.

Alignment of TpAP1 with other antimicrobial peptides reveals a high level of similarity. Im1 (ACD11790.1) and Im4 (C0HL59.1) from *Isometrus maculatus* are identical to TpAP1, whereas ToAP1 (A0A1D3IXR7.1) from *T. obscurus* and Im2 (ACD11789.1) from *I. maculatus* share 94.1% identity. The differences include a single amino acid substitution: I to F

Table 4. Sequence and TPM of peptide TpAP1. The bolded region indicates the signal peptide, the underlined area shows the propeptide, and shaded residues represent the C-terminal amidation motif.

| Name | Sequence | TPM |
|------|----------|-----|
| TpAP1 | **MQMKYLIPIFFLVLIVADHCHA**FIGMIPGLIGGLISA-IKGRRKRDITAQLEQYRNLQKREAEIEDLLANLPVY | 18,904,74 |

at position 16 in ToAP1 and I to V at position 8 in Im2. TsAP-2 (S6D3A7.1) from *T. serrulatus* shows 88.2% similarity, with substitutions at positions 2 (I→L) and 16 (I→F) (Fig 8).

The molecular structure features a random coil from residues 1–5, followed by an α-helix that extends to the C-terminal region (Fig 9). Ser14, found at the base of the alpha helix, has an exposed hydroxyl group that helps stabilize the structure and may participate in polar interactions. Lys17, located further along the same helix, presents its positively charged side chain, which is oriented for electrostatic recognition. Both residues, considering their location and chemical characteristics, are essential for functional interaction.

## Potassium channel modulator peptides

Out of the 45 transcripts identified as potassium channel modulator peptides, TpK8—ranked eighth in TPM—was selected for further analysis because it was previously partially identified as Tp10 [16]. The sequence includes a 22-amino-acid signal peptide and a 39-amino-acid mature peptide that contains eight cysteines, forming four disulfide bonds (Table 5). The mature peptide has a molecular mass of 4335.97 Da, is highly water-soluble at pH 7, has an isoelectric point of pH 10.22, and carries a net charge of +3.1.

TpK8 exhibited 97.4% similarity with Ts6 (P59936.3, *T. serrulatus*) and TtBut (P0C168.1, *Tityus trivittatus*), with differences only in the lack of a tryptophan at the N-terminal and a F to Y substitution at position 20. Additionally, Tcis17 (WDU65862.1) from *Tityus cisandinus* and Tco30 (P0C185.1) from *Tityus costatus* showed high similarity, with identity percentages of 90% and 94.9%, respectively (Fig 10). The most notable feature of the alignment is the preservation of cysteine (C) residues, which form four disulfide bridges (C1-C4, C2-C5, C3-C6, C7-C8) that create the ICK motif. Tx608 from *Buthus israelis* (B8XH42.1) and Im_KTx, and a hypothetical protein from *Isometrus maculatus* (ACD11762.1), exhibit approximately 52% sequence identity. While both share the ICK framework and a lysine residue in the pore-binding loop, they differ in adjacent residues, with TCSAQA versus KAEGRA in TpK8. The Tcis21, from *Tityus cisandinus* (WDU65866.1) and Ts7, from *Tityus serrulatus* (P46114.2), show approximately 45% identity, exhibit significant variation, with a tryptophan (W) or phenylalanine (F) residue at the start, and a different pattern of hydrophobic and charged residues.

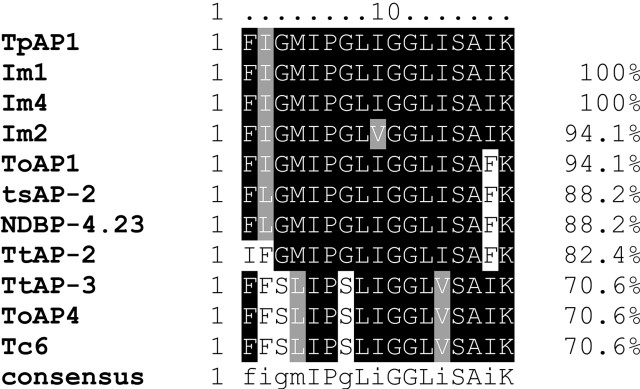

**Fig 8. Alignment of amino acid sequences of the putative antimicrobial peptide identified in the venom gland transcriptome of *Tityus paraguayensis* (TpAP1) with Im-4 (C0HL59.1), Im1 (ACD11790.1), and Im2 (ACD11789.1) from *Isometrus maculatus*, TtAP-2 (P0DRB6.1) and TtAP-3 from *Tityus trivittatus*, ToAp1 and ToAp4 (A0A1D3IXR7.1 and A0A1E1WVR9.1) from *Tityus obscurus*, NDBP-4.23 and Tc6 (Q5G8B5.1 and Q5G8B3.1) from *Tityus costatus*, and TsAP-2 (S6D3A7.1) from *Tityus serrulatus*.** Gray scale variations indicate sequence conservation. Uppercase letters in the consensus line show 100% conservation, lowercase letters denote the most conserved residues, blank spaces signify highly variable positions, and dashes (–) represent alignment gaps. The percentage values on the right indicate the similarity between TpAP1 of *Tityus paraguayensis* and the other sequences.

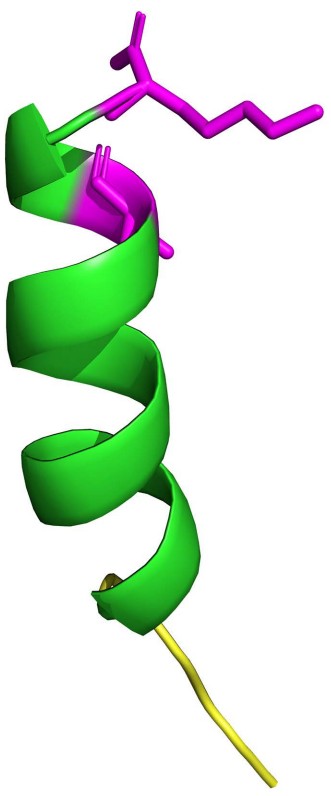

**Fig 9. Predicted three-dimensional structure of the TpAP1 peptide, generated through structural modeling with the AlphaFold server.** The image was rendered using PyMOL for illustration and shows a helical section of the toxin in green, with an adjacent loop in yellow, clearly indicating two polar residues in magenta.

**Table 5. Sequence and TPM of peptide TpK8. The bolded region indicates the signal peptide, and the shaded residues represent cysteine amino acids.**

| Name | Sequence | TPM |
|------|----------|-----|
| TpK8 | **MKVLCGLLLIFVLCSMIYLSEQ**CSTCLDLAC-GASRECYDPCYKAFGRAHGKCMNNKCRCYT | 907,04 |

Phylogenetic analysis shows that the TpK8 sequence clusters with toxins from mainly New World genera, including *Tityus*, *Rhopalurus*, and *Centruroides* (Fig 11). The close relation to *Tityus* species—especially *T. stigmurus*, *T. serrulatus*, and *T. cisandinus*—suggests a possible neotropical origin or adaptation. Species from *Mesobuthus* and *Leiurus* (Europe/Asia) form more basal branches, while *Parabuthus* (Africa) and other non-American Buthidae appear as external groups, reflecting adaptive radiation related to continental distribution. The inclusion of *Urodacus* (Australia) and *Grammostola* (spider) as outgroups highlights evolutionary and functional differences among lineages, indicating that TpK8 belongs to a distinct structural lineage of New World scorpion toxins.

Molecular modeling of TpK8 revealed an antiparallel β-sheet comprising two β-strands (β1: 28–31; β2: 34–37) and an α-helix spanning residues 12−23. Loop and random coil regions were also observed (Fig 12). The disulfide bonds stabilizing the structure were identified between cysteine residues 1–4, 19–37, 15–35, and 9–30. The TpK-8 peptide has a central core made up of two antiparallel β strands, connected by flexible loops to a stable α-helix on the right. Three key

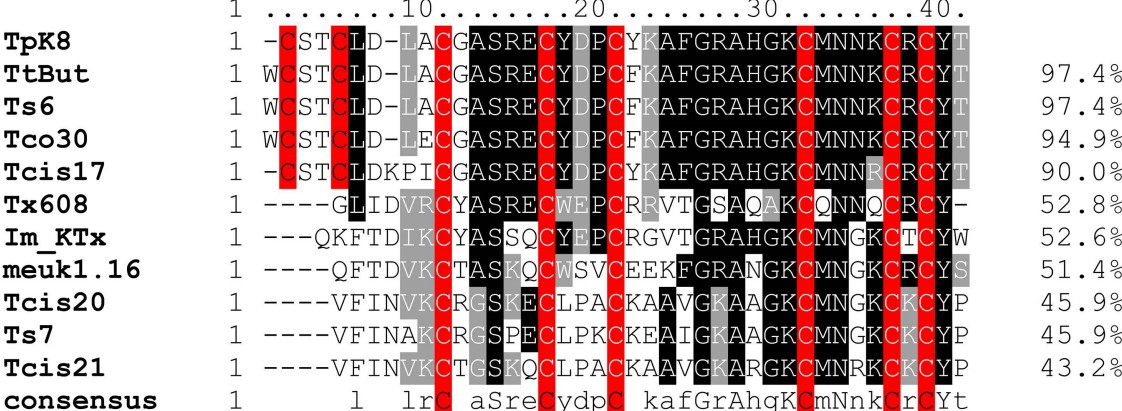

**Fig 10. Alignment of amino acid residues of KTx-type toxins identified in different scorpion species.** Sequences used include: Tcis17, Tcis20, and Tcis21 from *Tityus cisandinus* (WDU65862.1, WDU65865.1, and WDU65866.1), Ts6 and Ts7 from *Tityus serrulatus* (P59936.3 and P46114.2), TtBut from *Tityus trivittatus* (P0C168.1), Tco30 from *Tityus costatus* (P0C185.1), Tx608 from *Buthus israelis* (B8XH42.1), Meuk1.16 from *Mesobuthus eupeus* (AMY15321.1), and a hypothetical protein from *Isometrus maculatus* named Im KTx (ACD11762.1). Gray scale variations indicate sequence conservation. Uppercase letters in the consensus line represent 100% conservation, lowercase letters denote highly conserved residues, blank spaces mark highly variable positions, and dashes (–) indicate alignment gaps. The percentage values on the right indicate the similarity between TpK8 of *Tityus paraguayensis* and other sequences.

residues—magenta Phe24 (aromatic), Lys22 (positively charged), and Met29 (hydrophobic)—are strategically placed, while the pharmacological loop between the β strands is structurally designed to block Kv potassium channel pores.

We also compared the structure of the peptide modulators of potassium channels. TpK8 features the canonical ICK structure with a β-sheet core, an extended positively charged recognition loop, and key residues such as Lys27 (Fig 13A). TtBut, Ts6 and Tco30 (Fig 13B–13D) show nearly perfect backbone superposition, with conserved disulfide bridges, a hydrophobic core, and an identical pharmacological loop, including Lys27 and nearby residues like His28 and Asn29. The peptides Tcis17 and Tx608 (Fig 13E and 13F) both contain a conserved ICK motif, which confers structural stability. They include a basic residue (Lys or Arg) in the pore-binding loop. The electrostatic charge on the binding surface is less positively charged. Im-KTx, Meuk1.16, and Tcis20 retain only the ICK core, with a conserved basic structure (Fig 13G–13I). They have shorter loops with hydrophobic residues replacing charged ones. The active-site orientation prevents Lys27 from entering the channel pore. They have a neutral or negative electrostatic charge. Ts7 shares similarities with TpK8: both have a conserved ICK core, similar β-sheet topology, and disulfide bridges at matching positions (Fig 13J). Key differences include a shorter, modified recognition loop. Ts7's surface is more hydrophobic with side loops projecting differently. Tcis21 also resembles TpK8 in the ICK scaffold and disulfide bridges (Fig 13K), but differs in the pharmacological loop, which is compact, hydrophobic, and lacks the electropositive site, replaced by aromatic residues.

To investigate how TpK8 interacts with potassium channels, three channels were selected for molecular docking analysis. Fig 14 shows the structural details of TpK8 binding to the Kv1.2 channel, combining 2D analysis, 3D modeling, and interaction mapping at the active site. The LigPlot+ diagram (Fig 14A) displays non-covalent contacts that facilitate channel blocking, including hydrogen bonds between Gly24 (TpK8) and Gln357 (located in the pore helix of Kv1.2), Asn32 (TpK8) and Arg297 (positioned in the extracellular turret region), Thr39 (TpK8) and Ser360 (pore helix), and a key electrostatic interaction between Asp379 (vestibule extracellular domain) and Tyr16 (TpK8), along with several hydrophobic interactions involving Ala26 and nearby residues. Fig 14B presents a 3D view of the complex, with the Kv1.2 tetramer and TpK8 located in the extracellular vestibule. The orientation of the pharmacological loop toward the pore and surface complementarity highlights the detailed interaction architecture. In Fig 14C, the enlarged selectivity filter emphasizes essential blocking interactions, such as Ala26-Gln357 (pore helix), Gly24-Gln357 (pore helix), and the salt bridge Arg297

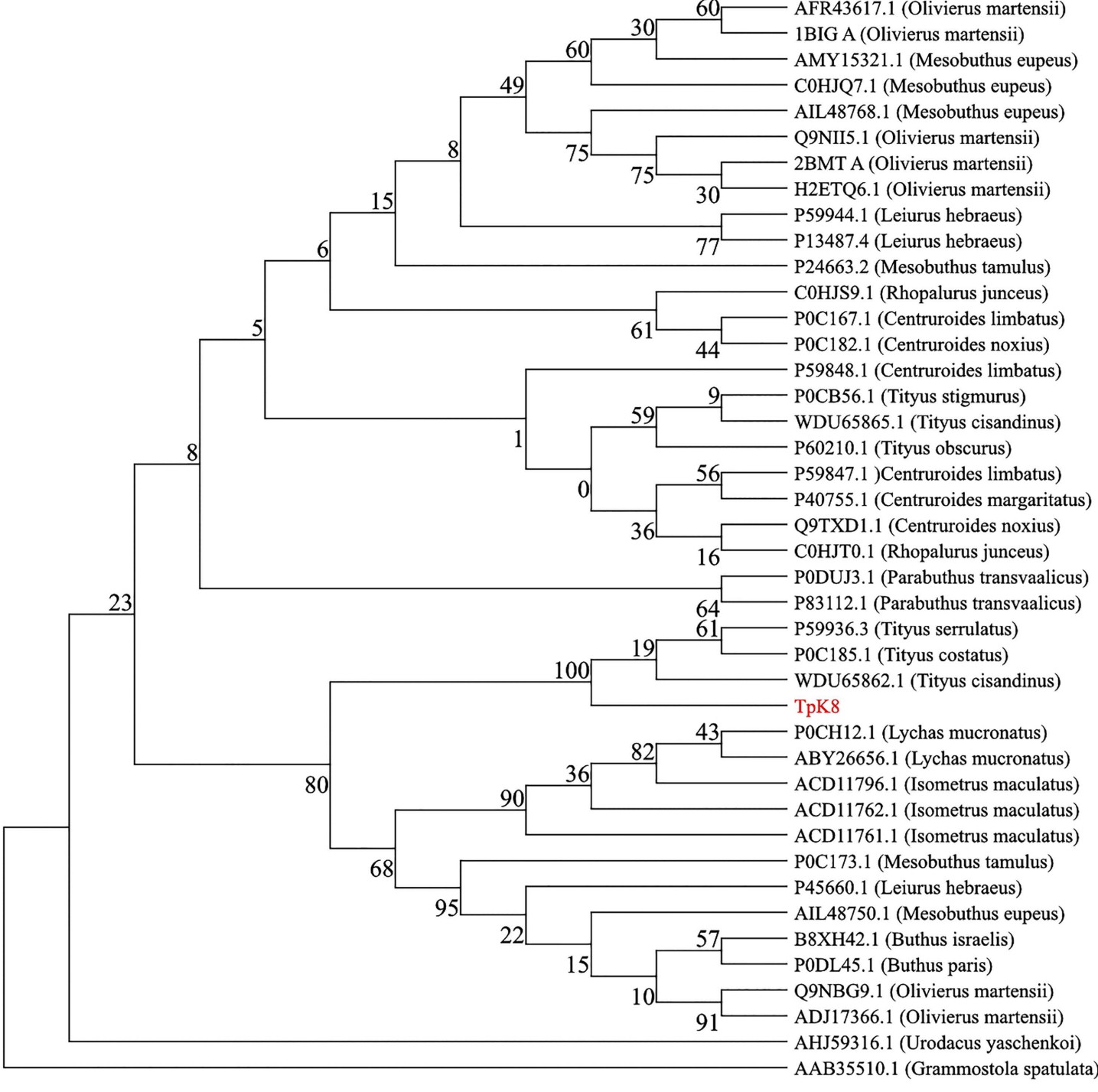

**Fig 11. The phylogenetic tree shows the evolutionary relationships of the TpK8 toxin (from *Tityus paraguayensis*, highlighted in red) with other arachnid toxins.** Most of the sequences analyzed are from the Buthidae family including: *Tityus cisandinus* (WDU65865.1, WDU65862.1), *T. stigmurus* (P0CB56.1), *T. obscurus* (P60210.1), *T. serrulatus* (P59936.3), *T. costatus* (P0C185.1); *Centruroides limbatus* (P0C167.1, P59848.1), *C. noxius* (P0C182.1, Q9TXD1.1), *C. margaritatus* (P40755.1); *Olivierus martensii* (AFR43617.1, 1BIG A, Q9NII5.1, 2BMT A, H2ETQ6.1, Q9NBG9.1, ADJ17366.1); *Mesobuthus eupeus* (AMY15321.1, C0HJQ7.1, AIL48768.1, AIL48750.1), *M. tamulus* (P24663.2, P0C173.1); *Leiurus hebraeus* (P59944.1, P13487.4, P45660.1); *Rhopalurus junceus* (C0HJS9.1, C0HJ10.1); *Parabuthus transvaalicus* (P0DUJ3.1, P83112.1); *Lychas mucronatus* (P0CH12.1, ABY26656.1); *Isometrus maculatus* (ACD11796.1, ACD11762.1); *Buthus israelis* (B8XH42.1) e *B. paris* (P0DL45.1). The analysis also incorporated *Urodacus yaschenkoi* (AHJ59316.1) from the Urodacidae family. The toxin from *Grammostola spatulata* (AAB35510.1), belonging to the family Theraphosidae (Spiders), served as the outgroup to root the tree.

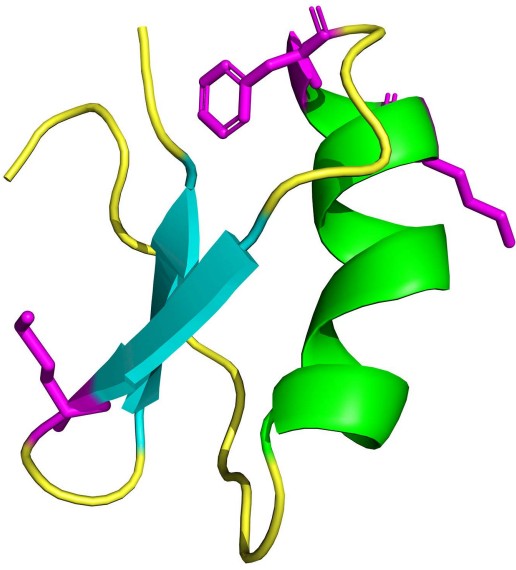

**Fig 12. Predicted three-dimensional structure of the TpK8 peptide, generated through structural modeling with the AlphaFold server.** The image was rendered using PyMOL for visualization. Ciano beta sheets, central region of the folded toxin nucleus. Green alpha helix located on the right. Yellow loops, including the pharmacological loop. Magenta highlights residues that are functionally important, such as Lys 22, Phe 24, and Met 29.

(turret region)-Asp32, with interatomic distances around 2.7–3.2 Å indicating strong contacts. These residues are in a hydrophilic environment that helps stabilize the toxin-channel complex. Additionally, according to HADDOCK parameters, the TpK8-Kv1.2 interaction model scores −79.4 ± 4.3 with a Z-score of −1.4, reflecting a favorable interaction. However, the high conformational variability (RMSD of 10.9 ± 0.5 Å) indicates flexibility within the complex.

The structural analysis of the TpK8–Kv1.3 complex reveals a two-dimensional interaction diagram created by LigPlot+ that illustrates the molecular network where key residues of TpK8—including Lys29, Tyr38, Thr39, His27, Asp17, and Ala26—form hydrogen bonds, hydrophobic contacts, and electrostatic interactions with residues of the Kv1.3 extracellular site, such as Ala465, Thr444, Gly446, Thr441, Lys458, and His451 (Fig 15A). Fig 15B shows a three-dimensional view of the complex, with the Kv1.3 channel in its tetrameric form and the TpK8 peptide, situated in the extracellular vestibule with its pharmacological loop facing the selectivity filter. Fig 15C emphasizes and enlarges key nearby interactions in the selectivity filter area, highlighting specific atomic contacts such as Lys29 with Thr444, Thr441 with His27, and Lys29 with Ala465—residues in critical regions like the selectivity filter (Thr441, Thr444), the P helix (Lys458, His451), and the extracellular vestibule (Ala465)—which support the blocking mechanism through structural and energetic compatibility. The molecular docking model indicates a strongly favorable and stable interaction, evidenced by a HADDOCK score of −101.4 ± 2.3, minimal conformational variability with an RMSD of 0.9 ± 0.5 Å, and solid statistical significance indicated by a Z-score of −2.1.

The molecular docking analysis presents a two-dimensional diagram of TpK8 and the Shaker channel interactions generated by LigPlot+ (Fig 16A). It highlights a network of non-covalent bonds involving residues Tyr16, Arg25, Gly24, and Asp17 of the peptide, which form specific contacts with Gly459, Tyr445, Thr449, and Lys456 of the channel. In Fig 16B, the three-dimensional model of the TpK8-Shaker complex shows the channel and peptide positioned in the extracellular vestibule, oriented favorably for interaction with the ion pore. Fig 16C illustrates the main proximal interactions near the selectivity filter, emphasizing atomic contacts such as Arg25 (TpK8) with Tyr445/Thr449 (channel's selectivity filter), Gly24 (TpK8) with Thr449 (selectivity filter), Tyr16 (TpK8) with Gly459 (P helix), and Asp17 (TpK8) with Lys456 (extracellular turret). These interactions include hydrogen bonds, hydrophobic forces, and a salt bridge, which stabilize the complex

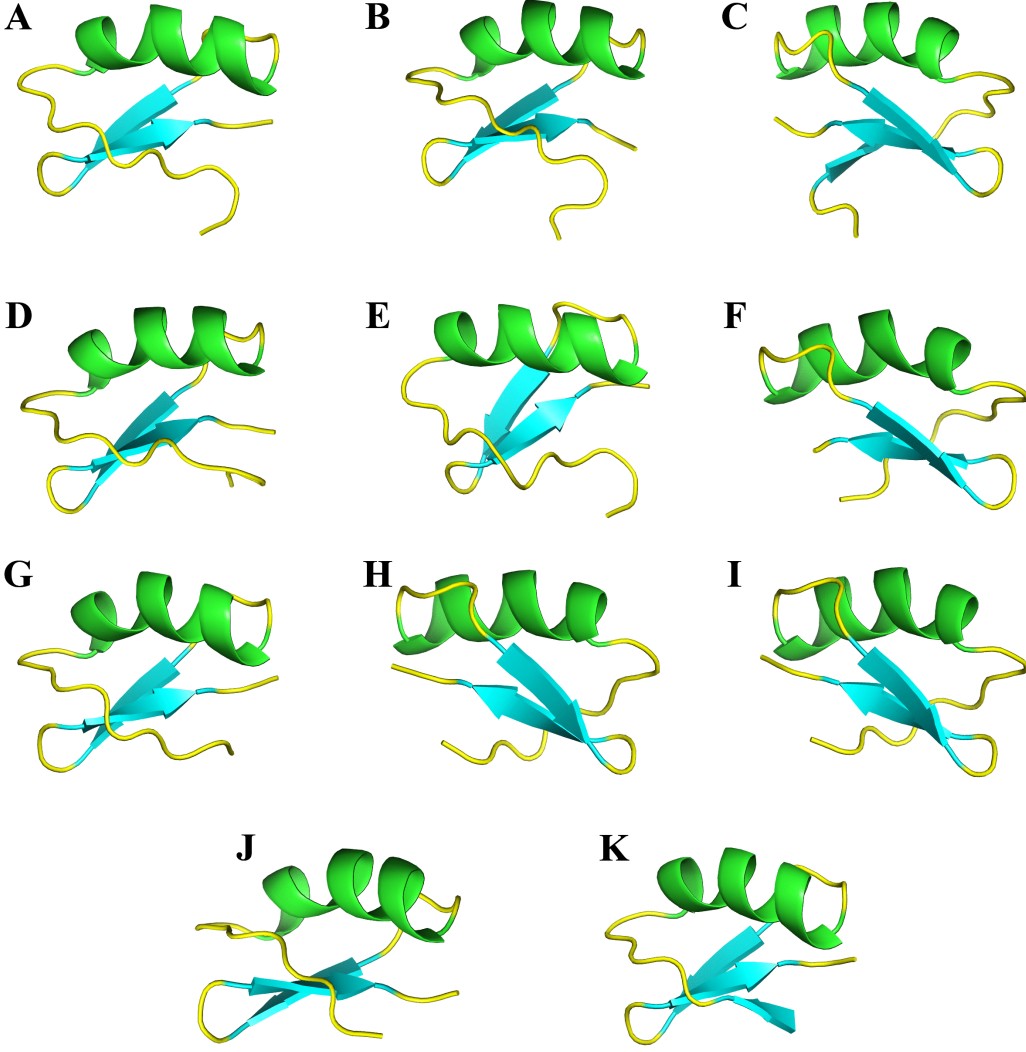

**Fig 13. Comparative analysis of the three-dimensional structure of TpK8 and homologous toxins.** The predicted three-dimensional structures of the TpK8 peptide and other toxins were generated using the AlphaFold server. **A.** TpK8 from *T. paraguayensis*. **B.** TtBut from *Tityus trivittatus*. **C.** Ts6 from *Tityus serrulatus.* **D.** Tco30 from *Tityus costatus*. **E.** Tcis17 from *Tityus cisandinus*. **F.** Tx608 from *Buthus Israelis*. **G.** A hypothetical protein from *Isometrus maculatus*, Im KTx. **H.** Meuk1.16 from *Mesobuthus eupeus*. **I.** Tcis20 from *Tityus cisandinus*. **J.** Ts7 from *Tityus serrulatus*. **K.** Tcis21 from *Tityus cisandinus*.

and facilitate channel function blocking. The TpK8–Shaker interaction model has a HADDOCK score of −76.0 ± 1.9 and a Z-score of −1.8, showing a favorable binding interaction. However, the high conformational variability (RMSD of 12.2 ± 0.1 Å) indicates considerable flexibility in the formed complex.

## Discussion

The venom profile and molecular traits of *T. paraguayensis's* toxins are unknown. Only one proteomic study identified some peptides and enzymes [16], leaving gaps in understanding *Tityus* toxin diversity and evolution. This is the first transcriptomic analysis of *T. paraguayensis* venom, revealing 523 transcripts linked to venom, mostly ion channel modulators, antimicrobial peptides, and antihypertensive agents. A key finding was the mismatch between transcript diversity

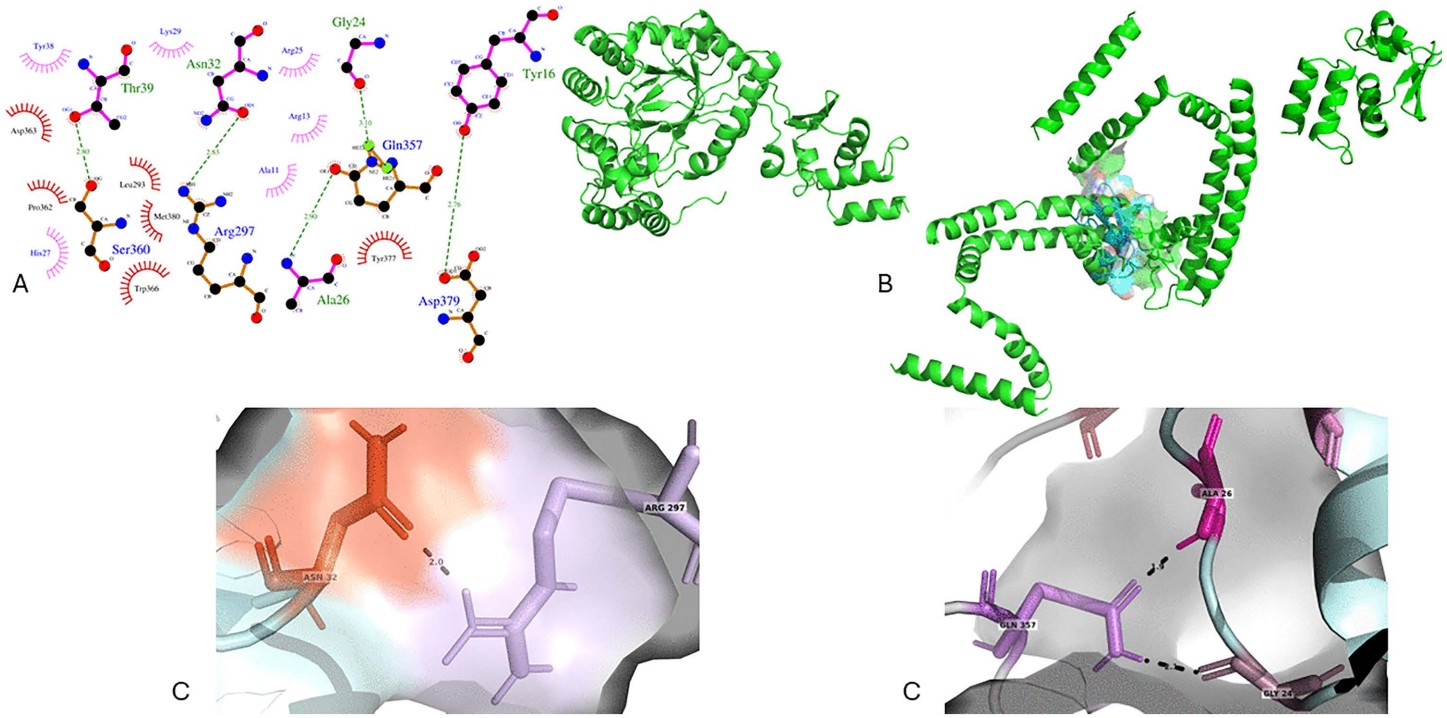

**Fig 14. Structural analysis of the TpK8-Kv1.2 interaction using molecular docking modeling. (A)** LigPlot+ produces a 2D diagram illustrating the interactions between TpK8 and the extracellular region of Kv1.2. **(B)** A 3D illustration of the TpK8–Kv1.2 complex, with the channel shown in green and the peptide in cyan. **(C)** An in-depth view of key intermolecular interactions, highlighting close contacts between the peptide and the channel's selectivity filter residues.

and abundance, with metalloproteases being diverse but lowly expressed. Four toxins were characterized: TpNa3 (sodium channel modulator), TpHyp1 (antihypertensive), TpAP1 (antimicrobial), and TpK8 (potassium channel modulator). In silico methods confirmed their identity and stability. TpK8 interaction with Kv1.2, Kv1.3, and Shaker channels showed stable blocking, especially for Kv1.3. The results reveal the toxin arsenal and bioactive compounds with potential for biotech tools and medicines.

The absence of the 28S subunit in RNA quality assessments is common in arthropods because cleavage during maturation produces two fragments (~1900 and 2000 nt) [33]. During RNA extraction and analysis, samples are heated, breaking hydrogen bonds that hold the 28S subunit's two fragments together. The 18S subunit, similar in size (1923 nucleotides), overlaps with these in electrophoresis [34]. This phenomenon has been observed in scorpions, spiders, centipedes, crustaceans, insects, and other arthropods [35].

Other studies also link the decline in sequencing read quality at the end to reaction products that hinder DNA polymerase activity [36,37]. Base composition bias is a contamination artifact caused by primers, common in Illumina sequencing protocols, due to random hexamer primers used in cDNA synthesis by reverse transcriptase [38]. BUSCO analysis compares genes in a genome or transcriptome with a reference set of orthologous genes to assess dataset completeness [39–41]. The observed difference in *T. paraguayensis* probably results from the reference dataset used, as this study utilized a metazoan gene database.

Our analysis revealed 523 potential venom components in the *Tityus paraguayensis* transcriptome, surpassing counts reported in several previous studies on related species. For instance, a transcriptome of *Tityus serrulatus* identified only 235 venom compounds from 5,282 assembled transcripts [42]. This notable difference probably arises from

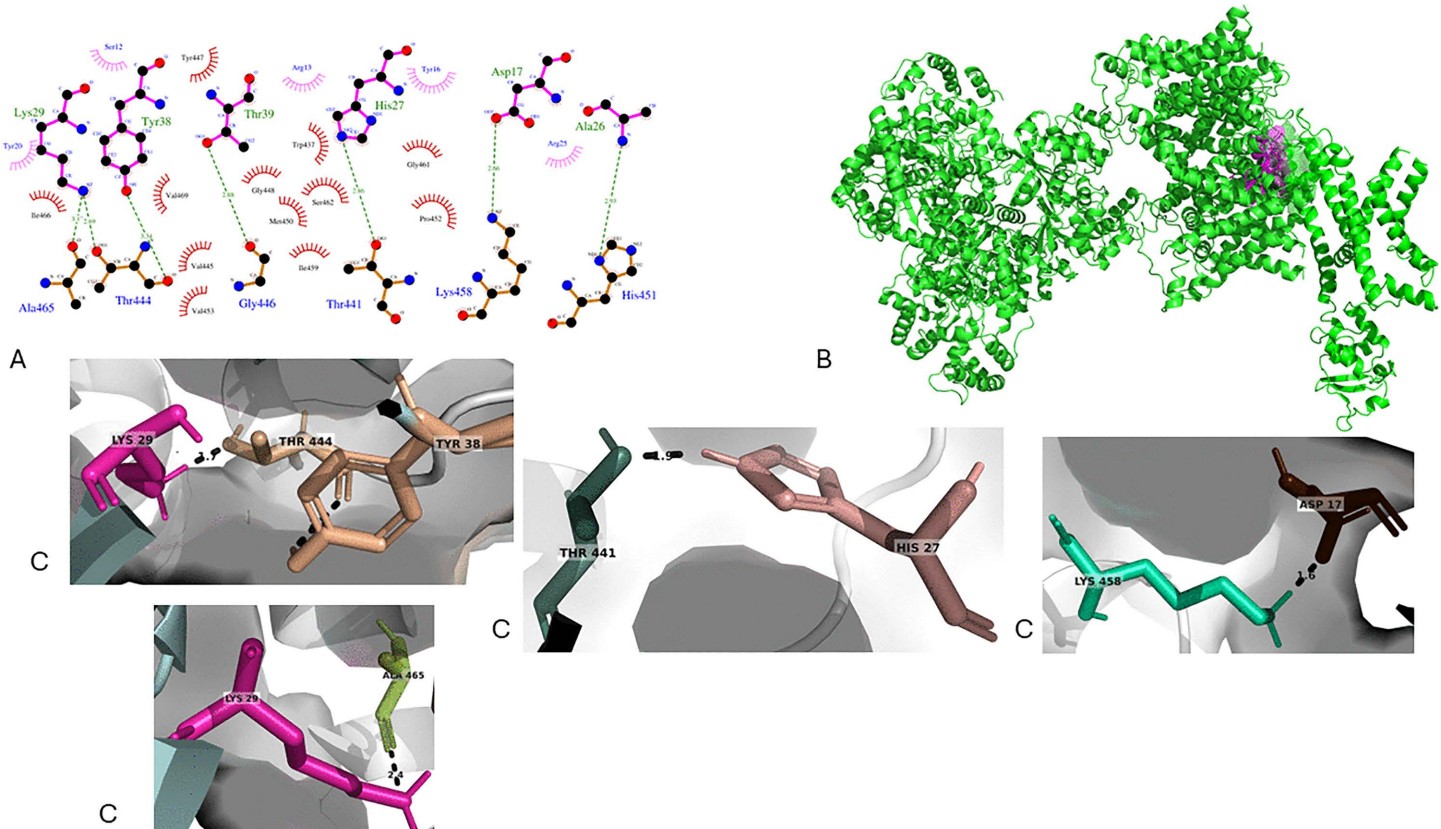

**Fig 15. Molecular docking model of the TpK8–Kv1.3 complex. A.** Two-dimensional diagram of the interactions between TpK8 and the extracellular site of Kv1.3 generated by LigPlot+. **B.** Three-dimensional view of the TpK8–Kv1.3 complex, with the channel in green and the peptide in magenta. **C.** Details of the main intermolecular interactions, emphasizing close contacts between the peptide and the channel's selectivity filter residues.

methodological variations, especially the use of advanced sequencing platforms and increased coverage here. This is supported by a study on *Megacormus gertschi*, which reported that Illumina sequencing yielded 110,538 transcripts and 182 venom-related components [43]. These comparisons show that although Illumina offers more comprehensive transcriptome coverage, the number of venom compounds identified primarily depends on sequencing depth and bioinformatic curation.

The diversity of venom compound classes we observed in *T. paraguayensis* aligns with the characteristic pattern documented for the Buthidae family. Consistent with other reports, enzymes—particularly proteases, such as metalloproteinases—emerged as the most diverse class in terms of distinct transcripts [13]. Ion channel modulators (targeting sodium, potassium, and calcium channels) constituted the second-most-diverse group. This finding correlates with their dominant abundance in the venom, as reflected in their high TPM values. Proteomic analysis confirmed this diversity, identifying specific toxin subtypes, including both α and β toxins for potassium (KTx) and sodium (NaTx) channels, as well as a metalloproteinase related to TsMS 2 from *T. serrulatus* [16]. Furthermore, non-disulfide-bridged peptides (NDBPs), including antimicrobial and antihypertensive peptides, accounted for a significant portion of transcript diversity [13]. This overall profile, where proteases and ion channel-active peptides make up most of the venom arsenal, is a conserved feature across the genus and has been similarly described in the transcriptomes of *T. serrulatus*, *T. obscurus*, and *T. bahiensis* [42,44].

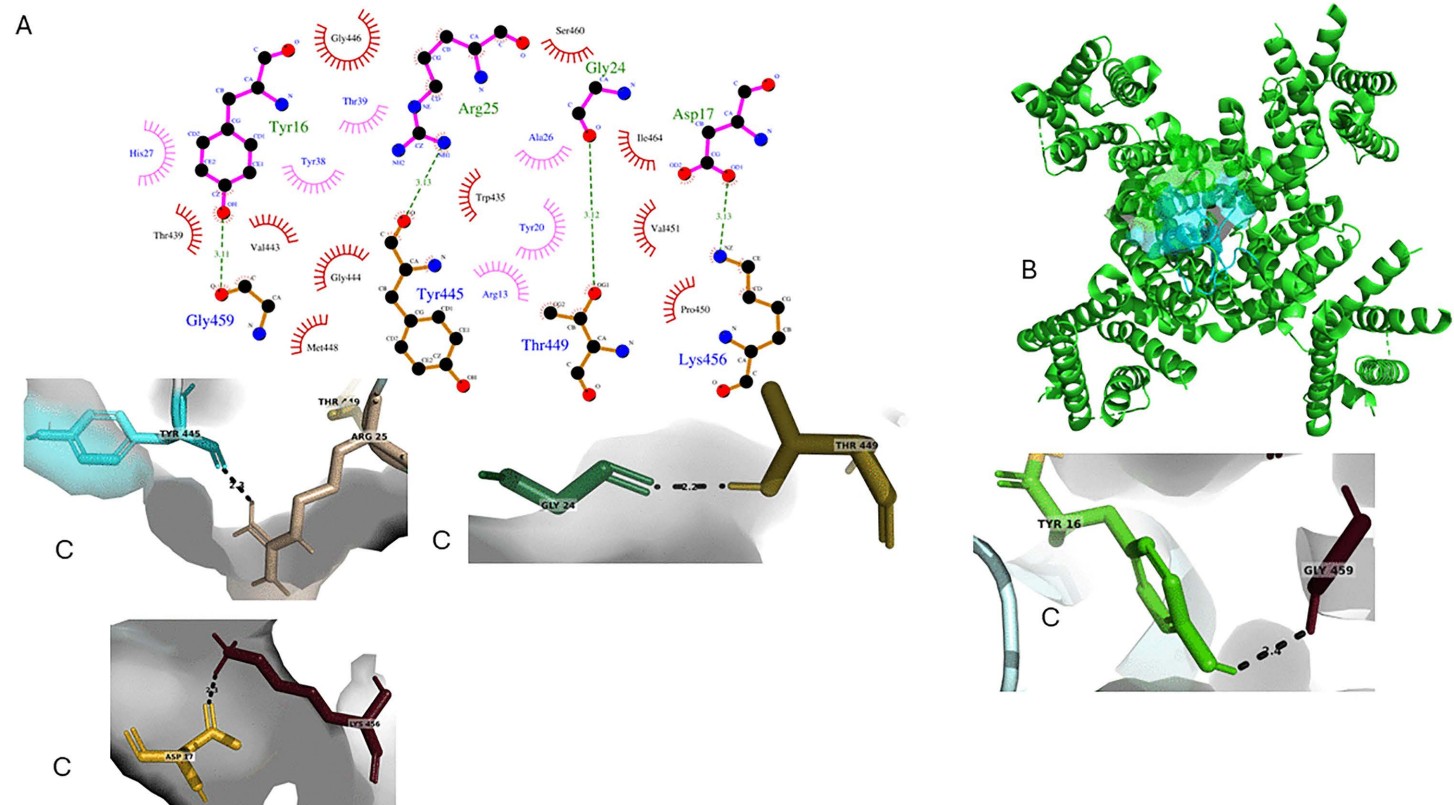

**Fig 16. Structural analysis of TpK8–Shaker interaction through molecular docking. A**. LigPlot+ generated a 2D diagram showing interactions between TpK8 and the Shaker extracellular site. **B**. 3D visualization of the TpK8–Shaker complex, with the channel in green and the peptide in magenta. **C**. Highlights of key intermolecular contacts, focusing on the close contacts between the peptide and the residues within the channel's selectivity filter.

The structural and phylogenetic evidence clearly suggests that TpNa3 belongs to the sodium channel β-toxin family. Its molecular mass and length match this group of peptides, which generally consist of 58–76 amino acids. Phylogenetic analysis reveals a close relationship with well-characterized β-toxins, such as To12 and Tf2, the latter of which is known for its selectivity for Nav1.3 channels [45,46]. This classification is further supported by the predicted three-dimensional structure of TpNa3, which displays the canonical βαββ motif—comprising an α-helix packed against an antiparallel β-sheet—that is a hallmark of β-toxins and is shared by other peptides like Cl13 from *Centruroides limpidus* [47], Toxin II from *Androctonus australis* [48], and BmK-βIT from *Mesobuthus martensii* [49]. This structural design of β-toxins binds to voltage-gated sodium channels and shifts their activation voltage to more negative values, promoting channel opening at resting membrane potential [50]. Therefore, TpNa3 likely operates through a similar mechanism, a hypothesis that requires further functional testing.

The characterization of hypotensive peptides in scorpion venoms remains limited, with confirmed members previously restricted to species like *Tityus serrulatus* and *T. stigmurus*. These peptides are typically defined by a short, mature sequence of 24–25 amino acids (≈2.7 kDa) and a conserved C-terminal KPP motif, which is critical for their function [51]. This family includes TsHpt-I, a potent B2 receptor agonist that exerts its antihypertensive effect through a mechanism distinct from that of bradykinin-potentiating peptides [52,53]. While TpHyp1 exhibits a considerable predicted size, its definitive classification as a hypotensive peptide is strongly supported by the presence of the characteristic KPP motif and a predicted α-helical structure, a feature also observed in the well-characterized TistH from *T. stigmurus* [54–56]. The

discrepancy in its predicted molecular mass likely stems from inaccuracies in the in silico identification of protease processing sites. This known challenge has also been reported in the transcriptomic analysis of *T. bahiensis* [42]. Therefore, TpHyp1 likely represents a new, longer variant of this peptide family, and its precise maturation process should be a focus of future proteomic studies.

The antimicrobial peptide TpAP1 identified in this study exhibits the typical features of non-disulfide-bridged peptides (NDBPs) from scorpion venoms, which are generally short, cationic, and amphipathic [57]. With a sequence of 17 amino acids and a modest net charge of +1, TpAP1 resembles pioneering peptides in this class, such as the 13-residue IsCT from *Opisthacanthus madagascariensis* [58,59]. Its predicted α-helical conformation, a hallmark of membrane-interacting AMPs, is remarkably similar to that of stigmurin from *T. stigmurus* [60]. Our structural model shows that this amphipathic helix is supported and precisely adjusted by key residues, including the polar Ser14 and the cationic Lys17, which are strategically located to facilitate interactions with microbial membranes [61]. The peptide's low net charge is an important trait, as a higher positive charge typically correlates with increased, and unwanted, hemolytic activity in host cells [62,63]. Therefore, the physicochemical profile of TpAP1—its short helical structure, specific functional residues, and low charge—indicates a mechanism targeting microbial membranes while suggesting minimal cytotoxicity, making it a promising candidate for the development of new anti-infective agents [64,65].

Building on the initial partial characterization of fraction Tp10 [16], this study successfully identified its complete sequence, recognizing it as the potassium channel toxin TpK8. This peptide, with 61 amino acids, falls within the typical size range for scorpion KTx peptides [66]. Phylogenetic analysis revealed that TpK8 is closely related to the α-KTx subfamily, showing high evolutionary similarity and up to 90% sequence identity with well-known toxins such as TtBut, Ts6, and Tco30, which block Kv1.2, Kv1.3, and Shaker channels [67]. The structural models emphasize the molecular basis of this activity, confirming the presence of the conserved inhibitory cystine knot (ICK) scaffold. Importantly, this scaffold displays the key functional motifs necessary for channel interaction: the "functional dyad" composed of Lys27 and a hydrophobic residue for pore blocking in Kv1 channels, and the critical Met29 residue, which interacts with the Shaker channel family [68]. Furthermore, the presence of Lys22, a residue shared with Ts6, suggests a possible specificity for the hERG channel, highlighting the multi-target potential of TpK8. [67,69]. The high structural similarity of TpK8 to its functional homologs, especially in the pharmacologically active loop, strongly supports its role as a potent and likely promiscuous modulator of potassium channels.

Molecular docking analyses provide a solid structural foundation for the high-affinity interactions between TpK8 and its potassium channel targets. The models demonstrated that TpK8 binds to the extracellular vestibule of the channels, with its pharmacologically active loop positioned directly above the selectivity filter. Notably, the interaction with Kv1.3 was remarkably stable, as shown by the highly favorable HADDOCK score of −101.4 ± 2.3 and low conformational variability (RMSD of 0.9 ± 0.5 Å) [70]. This strong binding was facilitated by key interactions, such as hydrogen bonds between TpK8 residues (Lys29, Thr39) and channel residues (Thr441, Thr444) within the selectivity filter, clearly illustrating the toxin's blocking mechanism. Conversely, the complexes with Kv1.2 and Shaker remained favorable but showed notably greater conformational flexibility (RMSD > 10 Å), indicating a more dynamic or lower-affinity interaction [71]. The detailed contact network, such as the salt bridge between Arg25 (TpK8) and Tyr445 (Shaker), corresponds to known recognition patterns of α-KTx toxins, in which a conserved lysine and matching residues fit into the channel pore [72,73]. These computational findings strongly indicate that Kv1.3 is a key high-affinity target for TpK8. This has important implications for using this toxin as a molecular probe or a potential therapeutic agent in autoimmune and inflammatory diseases where Kv1.3 is crucial [74].

The study's main limitations are its purely transcriptomic approach, which means that transcript (TPM) levels may not accurately reflect toxin protein levels due to post-transcriptional and post-translational regulation. Verifying the presence and quantification of toxins requires quantitative proteomics or targeted protein purification, with LC–MS/MS used to support quantification.. Toxin functions, such as sodium channel modulation, were mainly inferred from sequence homology,

without experimental validation using methods such as electrophysiology or antimicrobial testing. In silico modeling indicates that the predicted 3D structures and interactions require experimental validation using techniques such as X-ray crystallography or NMR. The analysis primarily focused on the most highly expressed toxins, overlooking low-abundance toxins like phospholipases (0.0001% TPM), which could still be biologically significant roles.

## Conclusions

523 putative toxin-related sequences were identified in the transcriptome of *T. paraguayensis* venom glands, including metalloproteinases, ion channel-modulating peptides, antimicrobial peptides, antihypertensives, and other compounds. These findings highlight the wide variety of compounds in *T. paraguayensis* venom, emphasizing their potential for pharmacological applications. Studies with the TpK8 toxin, including phylogenetic analyses, structural modeling, and molecular docking, demonstrated its high specificity and strong interaction with the Kv1.3 channel, making it a promising model for developing pharmacological and therapeutic tools. Future studies could explore the functional characterization, structure-activity relationships, and development of new therapeutics based on these venom components.

## Supporting information

**S1 Fig. Electropherogram showing the quality profile of total RNA extracted from the venom glands of *Tityus paraguayensis* scorpions.** The x-axis displays the run time in seconds, and the y-axis indicates fluorescence intensity measured by the equipment.
(TIF)

**S2 Fig. (A) Phred score for each sequenced base.** (B) Percentage content of each sequenced base: red/thymine; blue/cytosine; green/adenine; black/guanine.
(TIF)

**S3 Fig. (A) Phred score for each sequenced base.** (B) Percentage content of each sequenced base: red/thymine; blue/cytosine; green/adenine; black/guanine.
(TIF)

**S1 Table. Functional annotation.**
(DOCX)

## Author contributions

**Conceptualization:** Henrique Ranieri Covali-Pontes, Renata dos Santos Rodrigues, Malson Neilson Lucena.

**Data curation:** Jéssica de Moraes Carretone, Marcos Roberto Chiaratti, Milene Ferro, Malson Neilson Lucena.

**Formal analysis:** Henrique Ranieri Covali-Pontes, Brayhan Meneguelli, Milene Ferro, Malson Neilson Lucena.

**Funding acquisition:** Flavio Henrique Silva, Malson Neilson Lucena.

**Investigation:** Henrique Ranieri Covali-Pontes, Renata dos Santos Rodrigues.

**Methodology:** Henrique Ranieri Covali-Pontes, Brayhan Meneguelli, Jéssica de Moraes Carretone, Alynne Coelho Ribeiro, Angélica Camargo dos Santos, Thais Fernanda Carlos, Marcos Roberto Chiaratti, Milene Ferro, Flavio Henrique Silva.

**Project administration:** Malson Neilson Lucena.

**Resources:** Marcos Roberto Chiaratti, Milene Ferro, Flavio Henrique Silva, Renata dos Santos Rodrigues, Malson Neilson Lucena.

**Software:** Jéssica de Moraes Carretone, Milene Ferro.

**Supervision:** Flavio Henrique Silva, Malson Neilson Lucena.

**Validation:** Milene Ferro, Malson Neilson Lucena.

**Writing – original draft:** Henrique Ranieri Covali-Pontes, Brayhan Meneguelli, Alynne Coelho Ribeiro, Malson Neilson Lucena.

**Writing – review & editing:** Malson Neilson Lucena.

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
