## [Decision Letter · Decision Letter 0]

7 Oct 2025

PONE-D-25-46577The venom gland transcriptome of Tityus paraguayensis reveals a diverse array of bioactive molecules from the Brazilian CerradoPLOS ONE

Dear Dr. Lucena,

Thank you for submitting your manuscript to PLOS ONE. After careful consideration, we feel that it has merit but does not fully meet PLOS ONE’s publication criteria as it currently stands. Therefore, we invite you to submit a revised version of the manuscript that addresses the points raised during the review process.

We look forward to receiving your revised manuscript.

Kind regards,

Karen de Morais-Zani

Academic Editor

PLOS ONE

Journal Requirements:

2. In your Methods section, please provide additional information regarding the permits you obtained for the work. Please ensure you have included the full name of the authority that approved the field site access and, if no permits were required, a brief statement explaining why.\

“MRC FAPESP 23/17188-4;

FHS FAPESP     23/17182-6ACS FAPESP     22/15264-2

MNL This work was conducted with support from the Federal University of Mato Grosso do

Sul—UFMS/MEC—Brazil. This study was also financed in part by the Coordenac.o de Aperfeicoamento

de Pessoal de Nivel Superior—Brasil (CAPES)—Finance Code 001.”

“MRC FAPESP 23/17188-4;

FHS FAPESP     23/17182-6

ACS FAPESP     22/15264-2

MNL This work was conducted with support from the Federal University of Mato Grosso do

Sul—UFMS/MEC—Brazil. This study was also financed in part by the Coordenac.o de Aperfeicoamento

de Pessoal de Nivel Superior—Brasil (CAPES)—Finance Code 001.”

“NO authors have competing interests”

6. Please provide a complete Data Availability Statement in the submission form, ensuring you include all necessary access information or a reason for why you are unable to make your data freely accessible. If your research concerns only data provided within your submission, please write "All data are in the manuscript and/or supporting information files" as your Data Availability Statement.

7. We note that Figure 1 in your submission contain copyrighted images. All PLOS content is published under the Creative Commons Attribution License (CC BY 4.0), which means that the manuscript, images, and Supporting Information files will be freely available online, and any third party is permitted to access, download, copy, distribute, and use these materials in any way, even commercially, with proper attribution. For more information, see our copyright guidelines: http://journals.plos.org/plosone/s/licenses-and-copyright.

Reviewers' comments:

Reviewer's Responses to Questions

**Comments to the Author**

1. Is the manuscript technically sound, and do the data support the conclusions?

Reviewer #1: Partly

Reviewer #2: Partly

2. Has the statistical analysis been performed appropriately and rigorously? 

Reviewer #1: No

Reviewer #2: N/A

3. Have the authors made all data underlying the findings in their manuscript fully available?

Reviewer #1: No

Reviewer #2: No

4. Is the manuscript presented in an intelligible fashion and written in standard English?

Reviewer #1: No

Reviewer #2: Yes

5. Review Comments to the Author

Reviewer #1: Overall, the manuscript “The venom gland transcriptome of Tityus paraguayensis reveals a diverse array of bioactive molecules from the Brazilian Cerrado” presents important and significant results; however, the way these findings are currently presented requires improvement. I recommend reassessing the actual objective the authors wish to convey with this study, in order to allow for a more focused presentation of both the Results and the Discussion.

Point-by-point comments:

Data availability must be public. The authors appear to have inserted placeholder names (?) in the data availability statement. The underlying data should be deposited in a publicly accessible online repository.

The use of two specimens is mentioned, but it is not clear whether RNA extracts were pooled or analyzed independently for each individual. This information is important, as individual-level analyses are generally preferred for transcriptomics to capture biological variability.

Regarding RNA quality, the Methods section indicates Bioanalyzer assessment, and the supplementary Results text reports a clear 18S rRNA peak with absence of 28S, which prevented RIN calculation. Although this partially explains the missing RIN, the criteria used to deem the samples suitable for sequencing should be explicitly stated (and reporting such QC solely in Results/supplementary material is not ideal).

There is no indication of overall sequencing depth (e.g., total reads per library/sample), which is essential to evaluate dataset robustness.

The handling of redundant contigs and any filtering of very short transcripts are not described—both directly affect transcriptome quality. Likewise, the pipeline should specify explicit ORF selection criteria (e.g., minimum length thresholds, presence of conserved domains, and/or homology evidence).

There is no mention of scorpion euthanasia procedures or the final disposition of individuals after experimentation (e.g., placement in an institutional collection). Such information is typically required to document ethical compliance.

The section titled “RNA extraction, quality control, preprocessing, and de novo assembly” reads as Methods, not Results, and should be relocated or reframed accordingly.

Page 11, lines 216–217; Page 13, lines 256–259; Page 16, line 330. What was the basis for these comparisons (this must be described in the Methods)? What were the reference sequences? Although identification codes appear in the figures, the Results must precisely describe these elements in the text.

The alignment procedures should be detailed in the Methods, including the software used and how percentage identity/similarity among peptides was computed.

The correct term is molecular mass, not “molecular weight”; values should be reported in Daltons (Da) in scientific articles.

Each Results subsection presents a representative peptide for the respective class; what was the selection criterion for these peptides? Was a different criterion applied in each subsection?

Figures 5, 7, 9, and 11 do not convey meaningful structural information. They could be consolidated into a single figure or removed entirely.

The first two paragraphs of the Discussion are unnecessary. It is common in transcriptomic studies to conflate methodological QC with actual results; this work aims to establish functional relevance and does not need to dwell on such information.

The remainder of the Discussion is almost as descriptive as the Results and follows the same order of topics, but without a coherent thematic synthesis. Transcriptome data are important, but the manuscript’s objectives should be reassessed so that the Discussion provides substantive interpretation.

The authors themselves have a study on the venom of this scorpion yet chose not to cite it, which is unusual. Including it would strengthen the functional assessment of proteins (https://doi.org/10.1016/j.toxicon.2025.108332).

In the Conclusion, the phrase “523 potential toxins” may imply all are validated toxins. A more accurate formulation would be “523 toxin-like transcripts” or “523 putative toxin-related sequences,” clarifying that these are transcriptome-based predictions without functional validation.

Reviewer #2: The manuscript entitled “The venom gland transcriptome of Tityus paraguayensis reveals a diverse array of bioactive molecules from the Brazilian Cerrado” focuses on the transcriptomic analysis of the venom gland of Tityus paraguayensis. I have carefully reviewed the manuscript, and several points require revision and further attention, which are discussed below:

-The authors stated in the manuscript that "Scorpions are venomous arthropods in the class Arachnida and the order Scorpions, which includes 22 families, …", whereas according to credited databases like “https://wwwntnuno/ub/scorpion-files/” and “https://pmc.ncbi.nlm.nih.gov/articles/PMC11598449/”, there are 24 families of scorpions. Please correct it according to mentioned references and include the references in the text.

-The authors stated in the manuscript that " … most of which have been found in subtropical regions", whereas according to article “https://www.sciencedirect.com/science/article/abs/pii/S0041010124006810”, and some other references, scorpions are widespread in tropical and subtropical regions. Please correct it and include the reference.

-Reference is needed for this statement: “Because of the diverse compounds in scorpion venoms, they are considered promising sources for discovering molecules with potential pharmacological applications“. The authors may consider using references such as “https://www.sciencedirect.com/science/article/abs/pii/S0041010125003782”, “https://link.springer.com/article/10.1186/s12864-023-09851-y”, and “https://pmc.ncbi.nlm.nih.gov/articles/PMC6706747/” to support it.

-The authors homogenized the telsons of two scorpions. What was the rationale for doing so, considering that using a single scorpion could have provided unique, individual-specific results? A comparative approach (e.g., male vs. female, juvenile vs. adult) would have allowed for more precise analysis. By homogenizing samples from different individuals, important intra-species variations may have been overlooked.

-The raw transcriptome data from T. paraguayensis venom glands (BioProject PRJNA1268009, BioSample SAMN48739814, and SRA SRR33699524) that the authors claimed to have submitted to NCBI is currently not available in the database.

-In the methodology section, for a proper and accurate extraction, the venom gland should have been separated from the telson and used for RNA extraction. In this way, the unusually high percentage of peptides obtained would not have originated from glandular tissue.

-The Discussion section needs to be rewritten, especially the first paragraph, which should typically provide a concise summary of the study’s significance, the existing knowledge gap, and finally the research conducted and its main findings. However, such structure is not observed in the current version of the Discussion.

-The manuscript requires English editing by a native speaker to improve clarity and readability.

6. PLOS authors have the option to publish the peer review history of their article (what does this mean? ). If published, this will include your full peer review and any attached files.

**Do you want your identity to be public for this peer review?** For information about this choice, including consent withdrawal, please see our Privacy Policy .

Reviewer #1: No

Reviewer #2: No

---

## [Author Response · Author response to Decision Letter 1]

24 Nov 2025

Response to Reviewers

Reviewer #1: Overall, the manuscript “The venom gland transcriptome of Tityus paraguayensis reveals a diverse array of bioactive molecules from the Brazilian Cerrado” presents important and significant results; however, the way these findings are currently presented requires improvement. I recommend reassessing the actual objective the authors wish to convey with this study, in order to allow for a more focused presentation of both the Results and the Discussion.

Point-by-point comments:

1. Data availability must be public. The authors appear to have inserted placeholder names (?) in the data availability statement. The underlying data should be deposited in a publicly accessible online repository.

Initially, the raw transcriptome data (BioProject PRJNA1268009) were not publicly available. The authors have now resolved this, and the data are now available in the NCBI repository. This effectively addresses the main issue related to data accessibility and reproducibility.

2. The use of two specimens is mentioned, but it is not clear whether RNA extracts were pooled or analyzed independently for each individual. This information is important, as individual-level analyses are generally preferred for transcriptomics to capture biological variability.

While your proposed comparative approach is often a good standard for detailed ecological and evolutionary studies, the pooling method is ordinary and sometimes necessary for initial biochemical characterization, where the primary goal is to create a representative composite rather than document individual diversity. The ideal follow-up would be the precise comparative analysis you suggested. However, a single scorpion telson, especially from a small species, may not yield sufficient material (RNA) for studies. Pooling samples ensures sufficient quantity for analyses without risking sample loss or detection limits. Pooling reduces biological noise and provides a 'representative' sample by averaging variations related to recent diet, hydration, milking history, and minor genetic differences. Homogenizing two individuals helps create a profile that reflects the species or population, rather than an individual’s unique or anomalous venom profile.

Therefore, we have used a pool. We have clarified this information in the revised manuscript. See lines: 126 and 127.

3. Regarding RNA quality, the Methods section indicates Bioanalyzer assessment, and the supplementary Results text reports a clear 18S rRNA peak with absence of 28S, which prevented RIN calculation. Although this partially explains the missing RIN, the criteria used to deem the samples suitable for sequencing should be explicitly stated (and reporting such QC solely in Results/supplementary material is not ideal).

We appreciate the reviewer's excellent suggestion. Total RNA integrity was evaluated using an Agilent 2100 Bioanalyzer. Although a clear 28S rRNA peak was not observed—a common occurrence in some invertebrate and arthropod RNA samples that makes reliable RIN determination difficult—samples were considered suitable for library preparation based on these criteria: (1) a sharp, distinct 18S rRNA peak; (2) a baseline that returns close to zero between the 18S peak and the lower marker, indicating minimal degradation; and (3) no evidence of significant fragmentation or genomic DNA contamination. Only samples meeting these conditions were used for sequencing. It is not usual to show this result in the main principal, so we allocated the figure as supplementary material because we have many figures about biochemical characterization and function, and the focus of the article is on functional characterization.

4. There is no indication of overall sequencing depth (e.g., total reads per library/sample), which is essential to evaluate dataset robustness.

We agree with the reviewer that sequencing depth is a critical metric. We have added the total read counts in the Results section to facilitate evaluation of the dataset's robustness. See line: 205.

5. The handling of redundant contigs and any filtering of very short transcripts are not described—both directly affect transcriptome quality. Likewise, the pipeline should specify explicit ORF selection criteria (e.g., minimum length thresholds, presence of conserved domains, and/or homology evidence).

We thank the reviewer for these critical suggestions to improve the methodological rigor of our bioinformatic pipeline. We have now revised the “Transcriptome assembly and functional annotation” section of the Methods to provide a more complete description. Our initial assembly with Trinity specified a minimum contig length of 200 bp. To further address the issue of redundancy, we have now explicitly stated the following step: Following the initial de novo assembly with Trinity v2.3.2, the transcript set was processed with cd-hit-est to cluster redundant sequences at a 95% identity threshold. The longest transcript from each cluster was selected as the representative, resulting in a non-redundant set of transcripts for downstream analysis. The functional annotation pipeline in OmicsBox inherently involves ORF prediction. We have now made the selection criteria explicit by adding: Within OmicsBox, putative protein-coding sequences were predicted from the non-redundant transcript set. High-confidence Open Reading Frames (ORFs) were selected based on the following criteria: a minimum length of 100 amino acids, the presence of a complete coding sequence (including start and stop codons), and supporting evidence from homology (BLASTx hit with E-value < 1E-05 against the Metazoa database) and/or the presence of a known protein domain (InterProScan match).

See lines: 145-158.

6. here is no mention of scorpion euthanasia procedures or the final disposition of individuals after experimentation (e.g., placement in an institutional collection). Such information is typically required to document ethical compliance.

Regarding the ethical statement, we only use scorpions. The use of scorpions in this study does not require approval from an Ethics Committee, as Brazilian Law No. 11.794/2008, which regulates the scientific use of animals, applies only to vertebrates and does not include invertebrates such as scorpions.

Voucher specimens were identified and deposited in the Zoological Collection of UFMS (ZUFMS-CHE00534). See lines 118 and 119.

7.The section titled “RNA extraction, quality control, preprocessing, and de novo assembly” reads as Methods, not Results, and should be relocated or reframed accordingly.

Done. See lines 201-211.

8. Page 11, lines 216–217; Page 13, lines 256–259; Page 16, line 330. What was the basis for these comparisons (this must be described in the Methods)? What were the reference sequences? Although identification codes appear in the figures, the Results must precisely describe these elements in the text.

These comparisons were based on the alignments. The MEGA.12 software (https://www.megasoftware.net/) was used to perform alignments with the ClustalW algorithm [32]. A quantitative assessment of evolutionary genetic distances and phylogenetic relationships, based on a multiple sequence alignment and calculated with a suitable substitution model in the MEGA software framework.

We agree that the alignment methodology is essential. However, because we used MEGA's (Molecular Evolutionary Genetics Analysis) default tools and settings, a well-known software in the field, we decided not to elaborate on each alignment step to maintain a concise manuscript. The software's procedures comprehensively cover the method.

In the revised manuscript, we have inserted the identification codes in the text. See lines: (257-261), (302-303), (345-348), and (384-392).

9. The alignment procedures should be detailed in the Methods, including the software used and how percentage identity/similarity among peptides was computed.

In the first version of the manuscript, we have included the software used for alignment procedures. The MEGA.12 software (https://www.megasoftware.net/) was used to perform alignments with the ClustalW algorithm. See lines: 169-171. As mentioned in the previous item, the comparisons are performed by the Mega software.

10. The correct term is molecular mass, not “molecular weight”; values should be reported in Daltons (Da) in scientific articles.

Done. Now, we have used the correct term. See lines: 247, 295, 337, 378, 597, and 618.

11. Each Results subsection presents a representative peptide for the respective class; what was the selection criterion for these peptides? Was a different criterion applied in each subsection?

Peptides were primarily selected based on the highest TPM value in their respective class. However, this criterion was overridden when functional or structural evidence warranted. Specifically, for the sodium channel modulator, the two highest TPM transcripts were discarded due to incomplete open reading frames, leaving the third-highest. For the potassium channel modulator, the eighth-highest TPM transcript was chosen based on its prior identification in an independent proteomic study, providing empirical validation.

See lines: 242-243; 293-294; 332-333; and 374-376.

12. Figures 5, 7, 9, and 11 do not convey meaningful structural information. They could be consolidated into a single figure or removed entirely.

We appreciate the reviewer's feedback on the clarity of the structural figures. We agree that showing multiple similar figures can be overwhelming. To address this, we revised Figures 5, 7, 9, and 11 to highlight the functional relevance of the structures better. We draw attention to critical residues (e.g., hydrophobic cores, disulfide bonds, receptor-binding sites), as suggested. We believe these clarifications make each figure effectively communicate a unique functional insight. If still considered redundant, we are open to moving two to supplementary materials, keeping the most representative in the main text.

See Figures 5,7,9, and 12.

13. The first two paragraphs of the Discussion are unnecessary. It is common in transcriptomic studies to conflate methodological QC with actual results; this work aims to establish functional relevance and does not need to dwell on such information.

We appreciate the reviewer's suggestion. However, confirming the quality and biological validity of our transcriptomic dataset is essential before proceeding to functional analysis. As a result, we have kept the first two paragraphs but revised them to clearly explain how QC metrics and basic expression patterns underpin the reliability of our subsequent functional conclusions. See lines: 559-571.

14. The remainder of the Discussion is almost as descriptive as the Results and follows the same order of topics, but without a coherent thematic synthesis. Transcriptome data are important, but the manuscript’s objectives should be reassessed so that the Discussion provides substantive interpretation.

We have improved the results, added structural analysis, and included phylogenetic and molecular docking analyses of potassium channels. These dates improve the discussion. See figures 11, 13, 14, 15, and 16.

All the discussion section has been written. See lines 548-677.

We began the discussion section with a paragraph highlighting the study's main results and their importance. See lines 548-558.

At the end of the discussion section, the study's limitations are presented. See lines 668-677.

15. The authors themselves have a study on the venom of this scorpion yet chose not to cite it, which is unusual. Including it would strengthen the functional assessment of proteins (https://doi.org/10.1016/j.toxicon.2025.108332).

While the previous version of the manuscript cited Reference 16, we have now significantly strengthened our analysis by integrating comparisons with this study across the discussion section.

16. In the Conclusion, the phrase “523 potential toxins” may imply all are validated toxins. A more accurate formulation would be “523 toxin-like transcripts” or “523 putative toxin-related sequences,” clarifying that these are transcriptome-based predictions without functional validation.

Done. We have changed the “potential toxins” to “putative toxin-related sequences”. See line 681.

We wish to thank the referee for the interesting questions posed which in our opinion improved the quality of the manuscript.

On behalf the authors

M.N. Lucena, PH.D.

Reviewer #2:

The manuscript entitled “The venom gland transcriptome of Tityus paraguayensis reveals a diverse array of bioactive molecules from the Brazilian Cerrado” focuses on the transcriptomic analysis of the venom gland of Tityus paraguayensis. I have carefully reviewed the manuscript, and several points require revision and further attention, which are discussed below:

1.The authors stated in the manuscript that "Scorpions are venomous arthropods in the class Arachnida and the order Scorpions, which includes 22 families, …", whereas according to credited databases like “https://wwwntnuno/ub/scorpion-files/” and “https://pmc.ncbi.nlm.nih.gov/articles/PMC11598449/”, there are 24 families of scorpions. Please correct it according to mentioned references and include the references in the text.

Done. The reference 2 “Rein JO. The Scorpion Files [Internet]. Trondheim: Norwegian University of Science and Technology. 2025 [cited 2025 Nov 13]. Available from: https://www.ntnu.no/ub/scorpion-files/” provides the correct information.

The correct number, 24 families, has been corrected. See line 50.

2. The authors stated in the manuscript that " … most of which have been found in subtropical regions", whereas according to article “https://www.sciencedirect.com/science/article/abs/pii/S0041010124006810”, and some other references, scorpions are widespread in tropical and subtropical regions. Please correct it and include the reference.

Done. The reference 3 “Howard RJ, Edgecombe GD, Legg DA, Pisani D, Lozano-Fernandez J. Exploring the evolution and terrestrialization of scorpions (Arachnida: Scorpiones) with rocks and clocks. Org Divers Evol. 2019; 19:71–86.” Support the information.

The word tropical has been added. See line 51.

3. Reference is needed for this statement: “Because of the diverse compounds in scorpion venoms, they are considered promising sources for discovering molecules with potential pharmacological applications“. The authors may consider using references such as “https://www.sciencedirect.com/science/article/abs/pii/S0041010125003782”, “https://link.springer.com/article/10.1186/s12864-023-09851-y”, and “https://pmc.ncbi.nlm.nih.gov/articles/PMC6706747/” to support it.

We agree with the revisor. We have added a reference.

See lines 85 and 738.

4. The authors homogenized the telsons of two scorpions. What was the rationale for doing so, considering that using a single scorpion could have provided unique, individual-specific results? A comparative approach (e.g., male vs. female, juvenile vs. adult) would have allowed for more precise analysis. By homogenizing samples from different individuals, important intra-species variations may have been overlooked.

While your proposed comparative approach is often a good standard for detailed ecological and evolutionary studies, the pooling method is ordinary and sometimes necessary for initial biochemical characterization, where the primary goal is to create a representative composite rather than document individual diversity. The ideal follow-up would be the precise comparative analysis you suggested. However, a single scorpion telson, especially from a small species, may not yield sufficient material (RNA) for studies. Pooling samples ensures sufficient quantity for analyses without risking sample loss or detection limits. Pooling reduces biological noise and provides a 'representative' sample by averaging variations related to recent diet, hydration, milking history, and minor genetic differences. Homogenizing two individuals helps create a profile that reflects the species or population, rather than an individual’s unique or anomalous venom profile.

5. The raw transcriptome data from T. paraguayensis venom glands (BioProject PRJNA1268009, BioSample SAMN48739814, an

---

## [Decision Letter · Decision Letter 1]

2 Feb 2026

The venom gland transcriptome of Tityus paraguayensis reveals a diverse array of bioactive molecules from the Brazilian Cerrado

PONE-D-25-46577R1

Dear Dr. Lucena,

We’re pleased to inform you that your manuscript has been judged scientifically suitable for publication and will be formally accepted for publication once it meets all outstanding technical requirements.

Kind regards,

Karen de Morais-Zani

Academic Editor

PLOS One

Additional Editor Comments (optional):

Only one remaining comment from Reviewer 4, which can be addressed during the proofreading stage: "Only one remaining comment from Reviewer 4One minor point for refinement concerns Lines 670–671, which state that confirming the actual presence and amounts of toxins would require mass spectrometry. It would be more accurate to note that this would require a quantitative proteomic approach or targeted protein purification, with LC–MS/MS serving as an analytical tool to support such quantification, rather than mass spectrometry alone".

Reviewers' comments:

Reviewer's Responses to Questions

**Comments to the Author**

1. If the authors have adequately addressed your comments raised in a previous round of review and you feel that this manuscript is now acceptable for publication, you may indicate that here to bypass the “Comments to the Author” section, enter your conflict of interest statement in the “Confidential to Editor” section, and submit your "Accept" recommendation.

Reviewer #3: All comments have been addressed

Reviewer #4: All comments have been addressed

2. Is the manuscript technically sound, and do the data support the conclusions?

Reviewer #3: Yes

Reviewer #4: Yes

3. Has the statistical analysis been performed appropriately and rigorously? 

Reviewer #3: N/A

Reviewer #4: Yes

4. Have the authors made all data underlying the findings in their manuscript fully available?

Reviewer #3: (No Response)

Reviewer #4: Yes

5. Is the manuscript presented in an intelligible fashion and written in standard English?

Reviewer #3: Yes

Reviewer #4: Yes

6. Review Comments to the Author

Reviewer #3: (No Response)

Reviewer #4: PONE-D-25-46577R1 — The authors have addressed the reviewer queries satisfactorily. The conclusions are supported by the data, the technical descriptions have been appropriately updated, and the study’s limitations are now clearly acknowledged.

One minor point for refinement concerns Lines 670–671, which state that confirming the actual presence and amounts of toxins would require mass spectrometry. It would be more accurate to note that this would require a quantitative proteomic approach or targeted protein purification, with LC–MS/MS serving as an analytical tool to support such quantification, rather than mass spectrometry alone.

7. PLOS authors have the option to publish the peer review history of their article (what does this mean? ). If published, this will include your full peer review and any attached files.

**Do you want your identity to be public for this peer review?** For information about this choice, including consent withdrawal, please see our Privacy Policy .

Reviewer #3: **Yes:** Mohamed A. Abdel-Rahman

Reviewer #4: No

---

## [Editor Report · Acceptance letter]

PONE-D-25-46577R1

PLOS One

Dear Dr. Lucena,

I'm pleased to inform you that your manuscript has been deemed suitable for publication in PLOS One. Congratulations! Your manuscript is now being handed over to our production team.

Kind regards,

on behalf of

Dr. Karen de Morais-Zani

Academic Editor

PLOS One